# The effect of slab touchdown on anticrack arrest in propagation saw tests

Philipp L. Rosendahl[1], Johannes Schneider[1], Grégoire Bobillier[2], Florian Rheinschmidt[1], Bastian Bergfeld[2], Alec van Herwijnen[2], and Philipp Weißgraeber[3]

[1]Technical University of Darmstadt, Institute of Structural Mechanics and Design, Germany
[2]WSL Institute for Snow and Avalanche Research SLF, Davos, Switzerland
[3]University of Rostock, Chair of Lightweight Design, Germany

**Correspondence:** Philipp Weißgraeber (philipp.weissgraeber@uni-rostock.de)

**Abstract.** Understanding crack phenomena in the snowpack and their role in avalanche formation is imperative for hazard prediction and mitigation. Many studies have explored how structural properties of snow contribute to the initial instability of the snowpack, focusing particularly on failure initiation within weak snow layers and the onset of crack propagation. This work addresses the subsequent stage, the effect of slab touchdown after weak layer failure in mixed-mode loading (compressive anticrack (mode I) and shear (mode II) loading). Our results demonstrate that slab touchdown reduces the energy release rate, which can lead to crack arrest even under static conditions. This challenges the idea that only the dynamic properties of snow layers and spatial snowpack variations govern arrest, emphasizing instead the crucial role of mechanical interactions between the slab, weak layer, and base layer. By integrating these findings into the broader context of snowpack stability analysis, we contribute to a more nuanced understanding of avalanche initiation mechanisms. The analysis is provided in a comprehensive open-source model (https://github.com/2phi/weac).

to be submitted in

**Natural Hazards and Earth System Sciences**

## 1 Introduction

Understanding the stability and behavior of anticracks in weak snowpack layers, i.e., cracks whose faces move towards each other instead of apart, is critical for avalanche hazard evaluation (Schweizer et al., 2003; van Herwijnen and Jamieson, 2005; Heierli and Zaiser, 2008). The propagation saw test (PST) has proven to be a useful field test to assess the stability of such cracks as a precursor to avalanche release (van Herwijnen and Jamieson, 2005; Gauthier and Jamieson, 2006; Sigrist et al., 2006). PST results serve as a proxy for the potential of weak layer cracks to propagate over large areas (Simenhois and Birkeland, 2009; Ross and Jamieson, 2012).

The potential of a crack to expand over a large area is decisive for the size of avalanches and thus the corresponding danger level. The propagation potential and possible crack arrest has been subject to numerous investigations (Jamieson and Johnston, 1992; Reuter and Schweizer, 2012; Gauthier and Jamieson, 2006, 2008; Schweizer et al., 2014a, 2016; Birkeland et al., 2014). Experimental studies show that crack propagation behavior is greatly affected by the properties of the snowpack. Specifically, the thickness and density of the overlying slab increase its stiffness, thus favoring full propagation and accordingly large avalanches (van Herwijnen and Jamieson, 2007). Experimental data further reveals that after the onset of crack propagation, a crack may arrest again after a certain distance (Bair et al., 2014; Bergfeld et al., 2023b)). Numerical investigations underline the influence of snowpack properties and further emphasize the significant influence of weak layer properties for crack propagation potential (Gaume et al., 2015b; Bobillier et al., 2024).

Using appropriate mechanical models, it is possible to determine the weak layer fracture toughness by analyzing the critical crack length at which cracks first become unstable (Schweizer et al., 2011; van Herwijnen et al., 2016; Rosendahl et al., 2019a; Rosendahl and Weißgraeber, 2020a; Weißgraeber and Rosendahl, 2023). When cracks become unstable, they either fully propagate through the isolated PST block (END), trigger slab failure and arrest as a consequence (SF), or arrest without triggering slab failure (ARR) (Gauthier and Jamieson, 2006; Benedetti et al., 2019; Bergfeld et al., 2023a). To discuss the dynamic behavior of a slab and weak layer system Siron et al. (2023) have proposed a variational approach and discussed the systems properties in comparison to the classical dynamic solution of Heierli (2005).

As the crack length increases, the system undergoes different states. When cracks are short, the slab is partially supported by the intact part of the weak layer and partially unsupported. At a certain crack length, the outermost corner of the slab comes into contact with the collapsed weak layer. As the crack continues to grow, an increasing portion of the slab is supported by the collapsed weak layer, a decreasing portion is supported by the uncollapsed weak layer, and in between the slab is unsupported (see Figure 1). This unsupported slab length, sometimes called the bridging length, has been studied by Heierli et al. (2008), Bair et al. (2014) and Benedetti et al. (2019) by using simplified mechanical models of slab deformation. While these contributions provided insights to fundamental properties and allowed for the comparision to experimental results, e.g., Bergfeld et al. (2021, 2023b), they do not provide a comprehensive explanation of the physical mechanisms underlying crack propagation and arrest. This knowledge gap highlights the need for a mechanical model that incorporates slab touchdown and can explain the physical behavior of crack propagation or arrest. Such models are mainly focussed on arrest conditions at the PST scale but can be used to study slope scale behavior. Such analyses must then be complemented by analyses slab failure Gaume et al. (2015a), spatial variability Schweizer et al. (2008) and topography Gaume et al. (2019) that all control propagation, arrest and hence, the size of avalanche release.

Here, we propose a closed-form analytical model for the static response of all phases of propagation saw tests. For this purpose, we follow Weißgraeber and Rosendahl (2023) and model the slab as a beam composed of an arbitrary number of layers resting on an elastic foundation representing the weak layer. By incorporating the consideration of slab touchdown, we aim to provide an understanding of the micromechanical and energy processes that dictate the behavior of weak layer cracks under varying conditions, in particular with respect to crack arrest. The model provides deformations, stresses, and mixed-mode energy release rates of weak layer cracks.

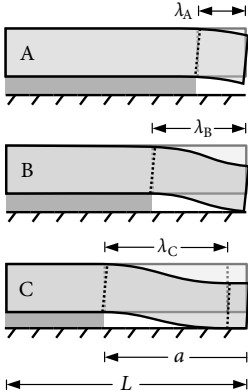

**Figure 1.** Three stages of a propagation saw test with total length $L$ and crack length $a$. For short cracks, the slab is partially unsupported and has a free end (A). The length of the unsupported segment is denoted $\lambda$ and, in this stage, equals the crack length $\lambda_A = a$. At a certain crack length, the end of the unsupported slab comes into contact with the collapsed weak layer (B). This adds support to the slab but allows it to rotate freely. The length of the unsupported segment and the crack length are equal $\lambda_B = a$. As the crack continues to grow, more of the slab is in contact with the collapsed weak layer (C). This adds an additional rotational constraint to the right end of the unsupported slab segment. The length of the unsupported segment is smaller than the crack length $\lambda_C < a$.

## 2 Methods

Slab touchdown in propagation saw tests represents a nonlinear mechanical contact problem because changing boundary conditions introduce structural nonlinearities into otherwise linear systems. By splitting the analysis of the different stages of a propagation saw test into subsystems of initially unknown length, the subsystems remain linear and closed-form solutions can be developed.

The present model builds on the foundational contributions of Heierli (2005), Benedetti et al. (2019), Siron et al. (2023), and Weißgraeber and Rosendahl (2023). It draws inspiration for the discrimination of PST crack propagation stages from Benedetti et al. (2019) and uses the framework introduced by Weißgraeber and Rosendahl (2023). We add the following advancements over the model of Benedetti et al. (2019): The present model employs first-order shear deformation theory to account for short beam sections, which invalidate the normality assumption of Euler-Bernoulli beam theory used in the Benedetti et al. (2019) model. Laminated plate theory is used to accurately represent slab behavior, capturing bending-extension coupling and the influence of slab layering on stiffness, not considered by Benedetti et al. (2019). Weak layer compliance, which was not considered in Benedetti's simplified approach, is modeled using a Winkler foundation and validated with finite element analyses. The employed weak-interface model identifies realistic shear and normal stress distributions in the weak layer, avoiding the oversimplified rigid beam segment assumption of the Benedetti et al. (2019) model. The present model calculates and uses energy release rates (ERR) to study brittle fracture phenomena in the weak layer, leveraging validated weak-interface concepts for ERR computation. Where Benedetti et al. (2019) base their analysis on finite crack-tip stresses that should be infinite at crack tips, the present study employs fracture mechanics concepts. They also proposed to use different

stages (referred to as schemes) but the stages did not yield continuous transitions between the stages, a limitation that we seek out to overcome.

Three different stages can occur in the considered system of the propagation saw test as shown in Figure 1. Stage A denotes short crack lengths where the slab under which the weak layer has been removed hangs freely. Stage B denotes conformations where cracks are long enough that the slab is in contact with the collapsed weak layer, but short enough that it is supported only at its outermost corner. At this stage, the slab is free to rotate at the point of contact. In stage C, the slab has a wider contact area with the collapsed weak layer, so that the part in contact can no longer rotate freely, but is restrained by the extended contact to the substratum.

## 2.1 Governing equations

Critical to the mechanical behavior of the touchdown problem are the deformations of the unsupported segment, whose length is called $\lambda$ (Figure 1). To model its structural response, we follow Weißgraeber and Rosendahl (2023) and describe the slab as a beam subject to the First-Order Shear Theory composed of an arbitrary number of layers by using theories developed for the analysis of layered composite materials Reddy (2003). The vertical deflection of the slab's midplane is given by

$$
\begin{aligned}
w_0(x) = c_1 + c_2 x + c_3 x^2 + c_4 x^3 \\
- \frac{A_{11}}{24(B_{11}^2 - A_{11}D_{11})} q x^4,
\end{aligned}
\tag{1}
$$

where $q$ is a line load representing the beam's weight and $A_{11}$, $B_{11}$, and $D_{11}$ are the beam's extension, bending–extension coupling, and bending stiffnesses, respectively (Weißgraeber and Rosendahl, 2023). Derivatives of the mid-plane deflection describe the cross-section rotation

$$
\psi(x) = \frac{B_{11}^2 - A_{11}D_{11}}{\kappa A_{55} A_{11}} w_0'''(x) - w_0'(x),
\tag{2}
$$

where $\kappa A_{55}$ is the beam's shear stiffness. The boundary conditions determining the constants $c_1$, $c_2$, $c_3$, and $c_4$ in Equation (1) are illustrated in Figure 2.

Reaction forces introduced by the slab resting on the intact weak layer (left end in Figure 2) are represented by linear and rotational springs with stiffnesses $k_\mathrm{N}^\circ$ and $k_\mathrm{R}^\circ$, respectively, where the superscript $\circ$ denotes the intact weak layer. With the coordinate origin at the segment's left end, the corresponding boundary conditions in all stages are

$$
M(x=0) = k_\mathrm{R}^\circ \, \psi(x=0),
\tag{3a}
$$

$$
V(x=0) = k_\mathrm{N}^\circ \, w_0(x=0),
\tag{3b}
$$

where $M$ and $V$ are bending moment and vertical section force, $\psi$ is the cross-sectional rotation of the slab, and $w$ is its vertical deflection. In stage A, the slab's right end is unsupported, i.e.,

$$
M(x=\lambda_\mathrm{A}) = V(x=\lambda_\mathrm{A}) = 0.
\tag{4}
$$

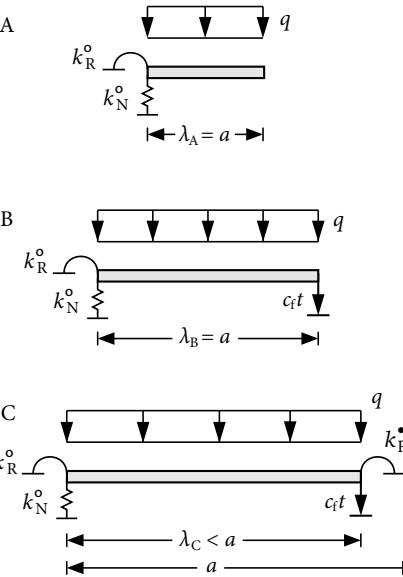

**Figure 2.** Boundary conditions of the unsupported segment. Reactions by the slab resting on the intact weak layer (○) are represented by linear and rotational springs with stiffnesses $k_N^\circ$ and $k_R^\circ$, respectively. Deflections are limited to a fraction $f_c t$ of the weak layer thickness. Moments introduced by the slab resting on the collapsed weak layer (●) are represented by rotational springs of stiffness $k_R^\bullet$.

In stage B, it is still free to rotate but vertical displacements are limited to the collapsed height of the weak layer. The fractional reduction of the weak layer thickness $t$ is expressed by the collapse factor $f_c \in [0, 1)$. The corresponding boundary conditions are

$$M(x = \lambda_B) = 0, \tag{5a}$$
$$w_0(x = \lambda_B) = f_c t. \tag{5b}$$

In stage C, the unsupported segment experiences additional reaction moments from the part of the slab that is in contact with the collapsed weak layer (right end in Figure 2). This is represented by a spring with rotational stiffness $k_R^\bullet$, where the superscript ● denotes the collapsed weak layer. The associated boundary conditions are

$$M(x = \lambda_C) = k_R^\bullet \psi(x = 0), \tag{6a}$$
$$w_0(x = \lambda_C) = f_c t. \tag{6b}$$

The magnitudes of the stiffnesses $k_N^\circ$, $k_R^\circ$, and $k_R^\bullet$ are determined using the principle of virtual forces. For this purpose, we apply a unit vertical force or a unit bending moment to a slab resting on an elastic foundation as modeled by Weißgraeber and Rosendahl (2023). The foundation represents the intact or collapsed weak layer. The elastic response is then captured by the

reciprocals of the corresponding displacements

$$k_{\mathrm{N}} = \bar{1}/w_0, \tag{7a}$$

$$k_{\mathrm{R}} = \bar{1}/\psi. \tag{7b}$$

where $\bar{1}$ denotes a virtual unit load.

## 2.2 Stage discrimination

The key to linearizing the otherwise nonlinear contact problem is knowing the crack lengths at which the system changes configurations from stage A to B and from stage B to C (Figure 1). The transition length $a_{\mathrm{AB}}$ between A and B corresponds to the crack length at which the maximum deflection of the slab is equal to the collapsed height of the weak layer. Solving Equation (1) for the boundary conditions of stage A, Equations (3) and (4), with the additional constraint $w_0(a_{\mathrm{AB}}) = f_{\mathrm{c}}t$, yields the polynomial of 4th order in $a_{\mathrm{AB}}$:

$$0 = \frac{a_{\mathrm{AB}}^4}{8K_0} + \frac{a_{\mathrm{AB}}^3}{2k_{\mathrm{R}}^\circ} + \frac{a_{\mathrm{AB}}^2}{2\kappa A_{55}} + \frac{a_{\mathrm{AB}}}{k_{\mathrm{N}}^\circ} - \frac{f_{\mathrm{c}}t}{q}, \tag{8}$$

where $K_0 = D_{11} - B_{11}^2 A_{11}^{-1}$. The smallest positive real root of Equation (8) is the transition length $a_{\mathrm{AB}}$. Only this transition from stage A to B matches the modeling approach by Benedetti et al. (2019) and their solution is contained as a special case of equation (8). The polynomial reduces to the contact length for first touchdown given by Benedetti et al. (2019), when following special case of the present model is assumed: the stiffness of the weak layer is infinitely high ($k_{\mathrm{N}}^\circ$, $k_{\mathrm{R}}^\circ \to \infty$), the shear stiffness of the slab is infinitely high ($\kappa A_{55} \to \infty$), the slab layering is symmetric (no bending-extension coupling ($B_{11} = 0$)), the slab is homogeneous without layering and the collapse height equals the weak layer thickness.

The transition between B and C occurs at $w_0'(a_{\mathrm{BC}}) = 0$, which marks the point at which the free end can no longer rotate freely and the contact area between slab and collapsed weak layer starts to increase. Using this condition together with the boundary conditions of stage B, Equations (3) and (5), yields a polynomial of 6th order in $a_{\mathrm{BC}}$

$$
\begin{aligned}
0 = {} & a_{\mathrm{BC}}^6 \, \kappa^2 A_{55}^2 \, k_{\mathrm{R}}^\circ k_{\mathrm{N}}^\circ \, q \\
& + a_{\mathrm{BC}}^5 \, K_0 \, \kappa^2 A_{55}^2 \, k_{\mathrm{N}}^\circ \, 6q \\
& + a_{\mathrm{BC}}^4 \, K_0 \, \kappa A_{55} \, k_{\mathrm{R}}^\circ k_{\mathrm{N}}^\circ \, 30q \\
& + a_{\mathrm{BC}}^3 \, K_0 \left( 2\kappa^2 A_{55}^2 \, k_{\mathrm{R}}^\circ + 3K_0 \, \kappa A_{55} \, k_{\mathrm{N}}^\circ \right) 24q \\
& + a_{\mathrm{BC}}^2 \, K_0 \left( q \, K_0 \left( \kappa A_{55} + k_{\mathrm{R}}^\circ k_{\mathrm{N}}^\circ \right) - \kappa A_{55} \right)^2 k_{\mathrm{R}}^\circ k_{\mathrm{N}}^\circ \, 72 \, f_{\mathrm{c}}t \\
& + a_{\mathrm{BC}} \, K_0^2 \, \kappa A_{55} \, 144 \left( k_{\mathrm{R}}^\circ q - \kappa A_{55} \, k_{\mathrm{N}}^\circ \, f_{\mathrm{c}}t \right) \\
& - K_0^2 \, \kappa A_{55} \, k_{\mathrm{R}}^\circ k_{\mathrm{N}}^\circ \, 144 \, f_{\mathrm{c}}t.
\end{aligned}
\tag{9}
$$

The smallest positive real root of Equation (9) corresponds to the transition length $a_{\mathrm{BC}}$.

While the length of the free segment $\lambda$ equals the crack length $a$ in stages A and B, in stage C its length $\lambda_\mathrm{C} < a$ is initially unknown. Using the boundary conditions of stage C, Equations (3) and (6), yields a polynomial of 6th order in $\lambda_\mathrm{C}$

$$
\begin{aligned}
0 = {} & \lambda_\mathrm{C}^6 \, \kappa^2 A_{55}^2 \, k_\mathrm{R}^\circ k_\mathrm{N}^\circ \, q \\
& + \lambda_\mathrm{C}^5 \, \kappa A_{55} \, k_\mathrm{N}^\circ \, (K_0 \, \kappa A_{55} + k_\mathrm{R}^\circ k_\mathrm{R}^\bullet) \, 6q \\
& + \lambda_\mathrm{C}^4 \, K_0 \, \kappa A_{55} \, k_\mathrm{N}^\circ \, (k_\mathrm{R}^\circ + k_\mathrm{R}^\bullet) \, 30q \\
& + \lambda_\mathrm{C}^3 \, K_0 \, \left( 2\kappa^2 A_{55}^2 \, k_\mathrm{R}^\circ + 3K_0 \, \kappa A_{55} \, k_\mathrm{N}^\circ + 3 k_\mathrm{R}^\circ k_\mathrm{N}^\circ k_\mathrm{R}^\bullet \right) 24q \\
& + \lambda_\mathrm{C}^2 \, 72 \, K_0 \, \Big( q \, K_0 \, \left( \kappa^2 A_{55}^2 + k_\mathrm{N}^\circ \, (k_\mathrm{R}^\circ + k_\mathrm{R}^\bullet) \right) \\
& \qquad\qquad + \kappa A_{55} \, k_\mathrm{R}^\circ \, (2 k_\mathrm{R}^\bullet q - \kappa A_{55} \, k_\mathrm{N}^\circ f_\mathrm{c} t) \Big) \\
& + \lambda_\mathrm{C} \, K_0 \, \kappa A_{55} \, 144 \, \Big( q \, K_0 \, (k_\mathrm{R}^\circ + k_\mathrm{R}^\bullet) \\
& \qquad\qquad - k_\mathrm{N}^\circ f_\mathrm{c} t \, (K_0 \, \kappa A_{55} + k_\mathrm{R}^\circ k_\mathrm{R}^\bullet) \Big) \\
& - K_0^2 \, \kappa A_{55} \, k_\mathrm{R}^\circ k_\mathrm{N}^\circ k_\mathrm{R}^\bullet \, 144 \, f_\mathrm{c} t .
\end{aligned}
\tag{10}
$$

Like before, the smallest positive real root of this equation provides the length of the unsupported segment $\lambda_\mathrm{C}$ in stage C.

### 2.3 Energy release rate

The elastic foundation representing the weak layer constitutes a so-called weak interface (Goland and Reissner, 1944; Lenci, 2001; Stein et al., 2015). In such weak interface models cracks are modeled by removing the elastic foundation. The corresponding energy release rate is given by

$$
\mathcal{G}(a) = -\frac{\partial \Pi(a)}{\partial a} = \frac{\sigma(a)^2}{2\,k_\mathrm{n}} + \frac{\tau(a)^2}{2\,k_\mathrm{t}},
\tag{11}
$$

where $a$ is the crack length, $\Pi(a)$ the total potential energy, $\sigma(a)$ and $\tau(a)$ are the weak layer's normal and shear stresses at the crack tip, and $k_\mathrm{n}$ and $k_\mathrm{t}$ are the weak layer's normal and shear stiffnesses (Krenk, 1992; Lenci, 2001). The shear and normal stresses are obtained within the framework introduced by Weißgraeber and Rosendahl (2023) by deformation analyses of the slab-weak layer system within the governing ODE system.

The consideration of energy release rates is crucial because, according to elastic continuum mechanics, stresses at sharp notches (such as V-notches, crack tips, anticrack tip) are theoretically infinite. However, many models yield finite stress values because they lack the necessary resolution to accurately capture the stress singularity at the crack tip. In weak-interface models, the weak layer of finite thickness is considered as a whole, and the crack itself is not explicitly resolved. In finite-element analyses, the stress at the notch tip increases as the element size decreases around the notch tip, regardless of how small the elements are chosen. This is due to the polynomial basis functions used in these elements. Similarly, discrete-element models employ elements of finite size whose geometries differ significantly from the microstructure of a weak layer. Inter-particle interactions are modeled using spring-type connections between the particles' centers of mass, and decreasing the particle size leads to increased observed stresses. Consequently, the stresses observed at crack tips are influenced by the chosen characteristic

**Table 1.** Slab layer nomenclature and properties. Slab properties are chosen in reference to three-layer slabs used by Habermann et al. (2008).

| Layer | Hand hardness index | Density $\rho$ (kg/m$^3$) | Young's modulus $E$ (MPa) | Poisson's ratio $\nu$ |
|---|---|---|---|---|
| Hard | P | 350 | 93.8 | 0.25 |
| Medium | 1F | 270 | 30.0 | 0.25 |
| Soft | 4F | 180 | 5.0 | 0.25 |
| Weak layer | F– | 100 | 0.3 | 0.25 |

**Table 2.** Dimensions and material properties.

| Property | Symbol | Value |
|---|---|---|
| Total length of PST blocks | $l$ | 5 m |
| Total slab thickness | $h$ | 36 cm |
| weak layer thickness | $t$ | 1 cm |
| Young's modulus weak layer | $E_{\mathrm{wl}}$ | 0.3 MPa |
| Young's modulus collapsed weak layer | $E_{\mathrm{c}}$ | 1.5 MPa |
| Poisson's ratio | $\nu$ | 0.25 |

length scale (element size, particle size, etc.). While the stress solutions provided by such weak interface models are generally correct, they are not accurate in the immediate vicinity of a notch. Therefore, it is not physically appropriate to use such stress values for stability assessments. However, a contour integral analysis Krenk (1992); Lenci (2001) gives a direct relation of the energy release rate of cracks within the weak layer with the finite maximal stress values within these models. To illustrate this, consider two weak layers with identical elastic properties (Young's modulus, Poisson's ratio) and the same cut length but different thicknesses. Although the crack-tip stresses in the thicker weak layer are smaller, the energy release rate of a crack in this weak layer is higher, making it the weaker configuration. This phenomenon has been demonstrated both theoretically and experimentally, for instance, in studies of adhesive layers, which share similar geometric and fracture characteristics with weak snow layers in terms of elastic contrast and failure localization (Stein et al., 2015).

Given the above limitations, the present approach allows for the computation of stresses in close agreement with computational methods such as FEA and DEM (see figure 5). However, because of their limited applicability in the analysis of fracture phenomena, we chose not to present details of their calculation. The interested reader is referred to Weißgraeber and Rosendahl (2023).

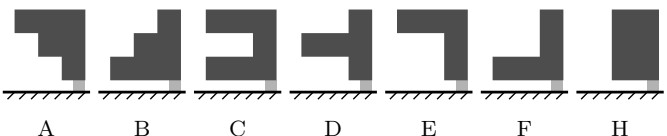

**Figure 3.** Benchmark snow profiles. Material properties of hard, medium, and soft slab layers (dark) and the weak layer (light) are given in Table 1. The weak layer is 1 cm thick and the slab layers have a thickness of 12 cm each.

## 3 Study design

### 3.1 Modeling propagation saw tests

We use slabs composed of three layers proposed as schematic hardness profiles by Schweizer and Wiesinger (2001), that are composed of soft, medium, and hard snow as benchmark slab configurations (Fig. 3). Assuming bonded slabs, we consider densities of $\rho = 350$, 270, and $180 \, \text{kg/m}^3$ for hard, medium, and soft snow layers with hand hardness indices pencil (P), one finger (1F), and four fingers (4F), respectively. From slab densities, we calculate the Young's modulus using the density-parametrization developed by Gerling et al. (2017) using acoustic wave propagation experiments and improved by Bergfeld et al. (2023b) using full-field displacement measurements

$$E_{\text{sl}}(\rho) = E_0 \left( \frac{\rho}{\rho_0} \right)^{\gamma}, \tag{12}$$

where $\gamma = 4.4$ controls density scaling (Bergfeld et al., 2023b), and $E_0 = 6.5 \cdot 10^3$ MPa and $\rho_0 = 917 \, \text{kg/m}^3$ are Young's modulus and density of ice (Northwood, 1947; Mellor and Cole, 1983; Moslet, 2007). Each slab layer is 12 cm thick and their individual material properties are given in Table 1. With reference to Jamieson and Schweizer (2000), who report weak layer thickness between 0.2 and 3 cm, we assume a weak layer thickness of $t = 1 \, \text{cm}$. Following density measurements of surface hoar layers by Föhn (2001) who reports densities i) between 44 and $215 \, \text{kg/m}^3$ with a mean of $102.5 \, \text{kg/m}^3$ and ii) between 75 and $252 \, \text{kg/m}^3$ with a mean of $132.4 \, \text{kg/m}^3$ using two different measurement techniques, we assume a weak layer density of $\rho_{\text{wl}} = 100 \, \text{kg/m}^3$, and a Young's modulus of $E_{\text{wl}} = 0.3 \, \text{MPa}$ (Bergfeld et al., 2023b). All parameters are summarized in Table 2.

### 3.2 Finite-element reference model

To validate the present model, in particular with respect to different slab layerings, we compare the present analytical solution to finite element analyses (FEA) using Abaqus 2020 (Simulia (Dassault Systèmes), France). The finite element model is assembled from individual layers with unit out-of-plane width. Each layer is discretized using at least 10 eight-node biquadratic plane-strain continuum elements with reduced integration through its thickness. The lowest layer corresponds to the weak layer. The lower edge of this layer is clamped, so that no nodal displacements or rotations can occur. weak layer cracks are introduced by removing all weak layer elements on the crack length $a$ over the thickness $f_c t$ and defining a node-to-surface contact between slab and the collapsed weak layer. The mesh is refined towards stress concentrations such as crack tips and mesh convergence

has been controlled carefully (Rosendahl and Weißgraeber, 2020a). The weight of the snowpack is introduced by providing the gravitational acceleration $g$ and assigning each layer its corresponding density $\rho$. Boundary conditions are free ends. In the FE model, the energy release rate of weak layer cracks

$$\mathcal{G}_{\mathrm{FE}}(a) = -\frac{\Pi(a + \Delta a) - \Pi(a - \Delta a)}{2\Delta a}, \tag{13}$$

is computed using the central difference quotient to approximate the first derivative of the total potential $\Pi$ with respect to $a$. The crack increment $\Delta a$ corresponds to the element size and could be increased twofold or threefold without impacting computed values of $\mathcal{G}_{\mathrm{FE}}(a)$.

### 3.3 Discrete-element reference model

We use a three-dimensional discrete-element-method (DEM) model in PFC3D (v5) (Itasca Software, USA) to validate the present model with respect to weak layer stresses and the influence of slope angle. The DEM model is established consists of three layers usually used in DEM models of weak layer slab systems (Bobillier et al. (2019)): a rigid basal layer, a transversely isotropic weak layer, similar to layers of surface hoar or facets, and a dense and uniform slab layer. The basal layer is composed of a single particle layer. The weak layer is created by cohesive ballistic deposition resulting in a porosity of $80\,\%$ with a layer thickness of $20\,\mathrm{mm}$. The slab layer is generated by cohesionless ballistic deposition and generated a uniform slab of $40\,\mathrm{cm}$ thickness. The slab has a macroscopic density $\rho = 250\,\mathrm{kg/m^3}$, a Young's modulus $E_{\mathrm{sl}} = 5.2\,\mathrm{MPa}$, and a Poisson ratio $\nu = 0.3$. The weak layer has a density $\rho = 110\,\mathrm{kg/m^3}$ and Young's modulus $E_{\mathrm{wl}} = 1\,\mathrm{MPa}$. The simulated PST is $7\,\mathrm{m}$ long and $30\,\mathrm{cm}$ wide. To evaluate the third touchdown stage (C), the weak layer is numerically removed over $3.5\,\mathrm{m}$ (crack length), creating a 'collapse' height of $20\,\mathrm{mm}$. Please refer to Bobillier et al. (2021) for a detailed description of the complete DEM PST model. The model setup intentionally differs from the FEA configuration to broaden the extent of the validation.

### 4 Results and discussion

Using the present analytical model and the numerical reference models (FEA and DEM) we investigate different configuration of propagation saw test experiments. We systematically study deformations, stresses, and mixed-mode energy release rates of weak layer cracks. The sensitivity regarding the governing parameters is studied and a comparison to experimental results is provided.

### 4.1 Slab layering and displacements

The model is used to study the deformations of PST configurations at different crack lengths. The benchmark snow profiles of layered slabs as shown in Figure 3 are used. Figure 4 shows propagation saw tests with $l = 5\,\mathrm{m}$ length, and different crack lengths $a = 0.9\,\mathrm{m}$, $1.6\,\mathrm{m}$, $3.3\,\mathrm{m}$ and $4.4\,\mathrm{m}$. Following the high-speed photography experiments by Van Herwijnen and Jamieson (2005) we assumed that the weak layer collapses to $50\,\%$ of its initial height, i.e., $f_{\mathrm{c}} = 0.5$. The solution of the present modeled are compared to FEA results for comparison.

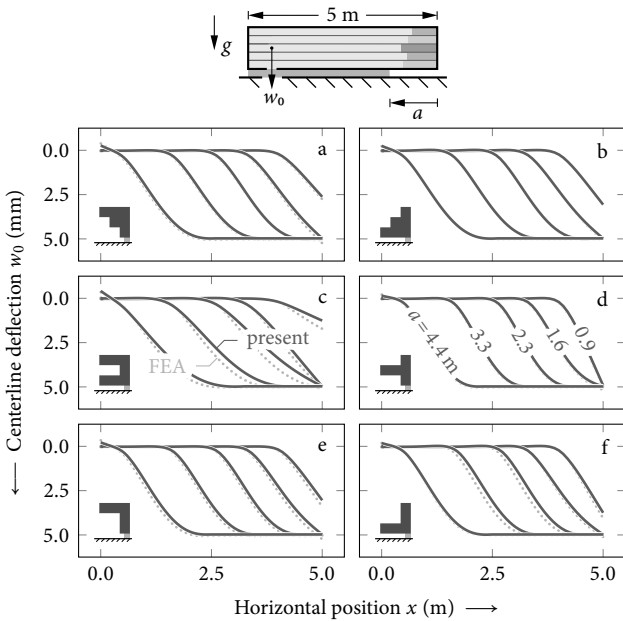

**Figure 4.** Deformations within PST configurations for the considered benchmark snow profiles (Figure 3) for different crack lengths ($a = 0.9\,\text{m}$, $1.6\,\text{m}$, $3.3\,\text{m}$ and $4.4\,\text{m}$). The deflections $w_0$ as obtained from the present model (solid lines) and the FEA reference solution (dotted lines) are shown. The weak layer is assumed to collapse to $50\,\%$ of its initial thickness ($f_c = 0.5$).

When $a = 0.9\,\text{m}$, all profiles are in stage A and none of the slabs are in contact with the collapsed weak layer (Figure 4a–f). Hence, we observe a free bending deformation of the slabs with the different profiles. At a crack length of $a = 1.6\,\text{m}$, touchdown is observed in all profiles ($w = f_c\,t$). However, the results show that their right ends can still freely rotate ($w_0' > 0$), indicating that they are in stage B (Figure 4a–f). With a further increase of the crack length to $a = 2.3\,\text{m}$, only profile C, which has the highest bending stiffness, remains in stage B (Figure 4c); all other profiles are in stage C (Figure 4a,b,d–f). At $a = 4.4\,\text{m}$, the crack is almost as long as the total length of the PST. Here, we observed negative (lifting) displacements ($w < 0$) at the left end (Figure 4a–f).

The results show a pronounced effect of the different snow profiles on the deformations in the considered PST configurations. Profiles with lower bending stiffness have higher bending induced inclinations than profiles with high bending stiffness, c.f. compare profile C and profile D in Figure 4. The change in the deformation behavior for different layerings also has an effect on the energy release rate. This has been discussed in detail in Weißgraeber and Rosendahl (2023), where also the effects of changed bending stiffness and the different mean densities are discussed in comparison. It is in line with field observations of increased snowpack stability when a pronounced bridging was observed, which is governed by slab bending and shear stiffness (Schweizer and Jamieson, 2003; Thumlert and Jamieson, 2014).

The findings reveal that snow profile properties, particularly bending stiffness, strongly influence PST deformation stages and crack propagation. Profiles with higher stiffness transition more gradually through stages, while lower stiffness profiles

show earlier transitions. These variations impact energy release rates and align with field observations linking bridging effects to increased snowpack stability. See section 4.7 for a discussion of the practical relevance on PST results.

## 4.2 Slope angle and weak layer stresses

In order to analyze the influence of the slope angle on weak layer stresses and slab displacements, we study different slope angles and compare the present model to the reference discrete-element model (DEM) described in section 3.3. To broaden the validation the models assumption differ from those of the previous FEA and we assume a homogeneous slab with macroscopic density $\rho = 250.0 \, \text{kg/m}^3$, Young's modulus $E_{\text{sl}} = 5.2 \, \text{MPa}$, Poisson's ratio $\nu = 0.3$, thickness $h = 40 \, \text{cm}$. In addition, a weak layer with Young's modulus $E_{\text{wl}} = 1.0 \, \text{MPa}$, thickness $t = 20 \, \text{mm}$, and collapse height $f_c = 1.0$ is assumed. The geometry of the PST is defined by its total column length $L = 7 \, \text{m}$, a unit out-of-plane width and the crack length $a = 3.5 \, \text{m}$.

At the considered crack length, touchdown of the slab occurs despite the high collapse height factor of $f_c = 1.0$. As the free end of the slab still shows an inclination for all three considered angles, all solutions are in stage B, see Figure 5. The model also shows the effect of cut direction within a PST (upslope vs. downslope) that changes the magnitudes of the shear and normal stresses for the cases of $\varphi = -35°$ and $\varphi = +35°$, as it is discussed, e.g., by Weißgraeber and Rosendahl (2023). Such insights allow to develop variations or improvements of propagation saw test experiments to improve the experimental approach or to derive guidelines for effective experiments (van Herwijnen et al., 2016; Birkeland et al., 2019; Bergfeld et al., 2023a).

The influence of slope angle on weak layer stresses and slab displacements reveals that touchdown occurs even with a high collapse height factor, and all configurations remain in stage B for the analyzed crack length. Variations in slope angle and cut direction (upslope vs. downslope) notably alter the magnitudes of shear and normal stresses, as highlighted in previous studies. These findings offer a foundation for refining propagation saw test experiments and developing guidelines to enhance their effectiveness.

## 4.3 Effect of crack length

The transition from stage A to B to C is directly linked with the displacement field of the slab. The rotation of the left free end caused by slab bending changes before (stage A) and during its first touchdown (stage B). When more and more of the slab comes in contact with the collapsed weak layer, the boundary on the left side becomes vertical (stage C). The progression of the displacement field of a homogeneous slab observed for increasing crack lengths $a$ is shown in Figure 6a.

Crack-tip loading increases with increasing crack length, which leads to a progressively increasing ERR in stage A ($a < 1.2 \, \text{m}$). Initial contact with the collapsed weak layer at the transition from stage A to B causes a sharp decrease in ERR ($a \approx 1.2 \, \text{m}$). This is caused by a reduced loading of the crack tip owing to the additional support of the slab. In stage B, the effect of added support remains present when the free end that rotated due to slab bending becomes vertical as the crack length increases. When the curvature of the unsupported segment changes its sign at the contact point, longer cracks cause an increase in ERR once again. At stage C the slab has developed its full contact with the base and its unsupported length no longer increases ($a > 2.3 \, \text{m}$). Therefore, the system reaches a steady state with a constant ERR until boundary effects at the right end

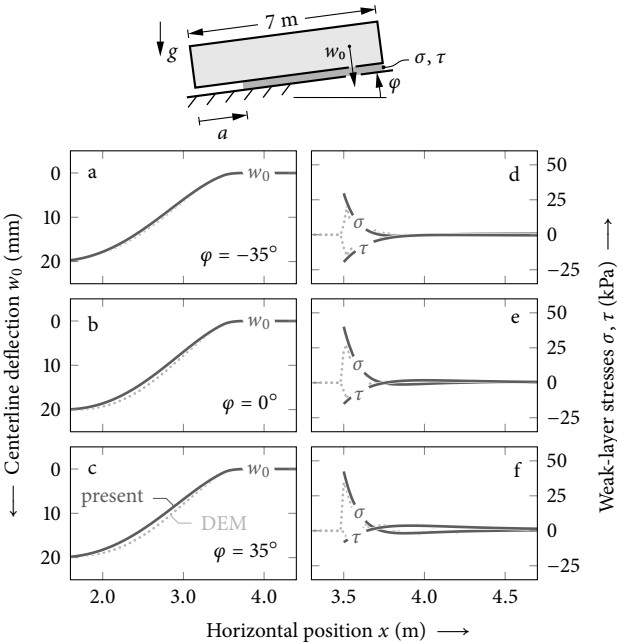

**Figure 5.** Model validation. Comparison of PST centerline deflections $w_0$, weak layer normal stresses $\sigma$, and weak layer shear stress $\tau$ between present model (solid) and DEM reference solution (dotted). PSTs with homogeneous slabs H (Figure 3) of $7\,\mathrm{m}$ total length with $a = 3.5\,\mathrm{m}$ cuts are shown at slope angles $\varphi = -35°$, $0°$ and $35°$, where negative slope angles are used to render down-slope cuts. The weak layer is assumed to collapse completely ($f_\mathrm{c} = 1.0$) from its initial thickness $t = 20\,\mathrm{mm}$.

become relevant. The evolution of the energy release rate (ERR) of weak layer cracks is shown in Figure 6c. The effect of

boundaries on the ERR of PST was also discussed by Bair et al. (2014) within a numerical FEA study. In their discussion of propagation saw test setups, they found that close ($0.5\,\mathrm{m}$ to $1.0\,\mathrm{m}$) to the end of a PST the ERR increased strongly, as also shown by the present model.

During stage A and B the unsupported length corresponds to the crack length. In stage C the effect of the touchdown, acting as a rotational constraint, reduces the unsupported length to less than the crack length. After the transition from stage B to stage

C the unsupported length is almost constant until the rotational constraint at the uncut end of the PST reduces and leads to shorter unsupported lengths. Figure 6b shows the corresponding unsupported length $\Lambda$ during the increase of the crack length in the PST.

The progression through stages A, B, and C illustrates the intricate relationship between slab displacement, crack-tip loading, and energy release rate (ERR). The unsupported length, directly tied to the crack length in stages A and B, decreases in stage

C as touchdown imposes a rotational constraint. This transition drives notable changes in ERR: an initial increase in stage A, a sharp drop at the onset of touchdown in stage B, and a steady state in stage C until boundary effects near the PST end cause

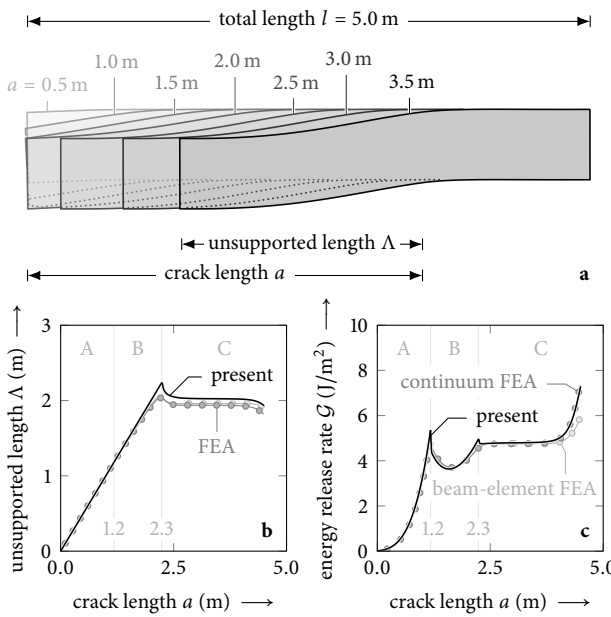

**Figure 6.** Evolution of the displacement field of a homogeneous slab in a propagation saw test with increasing crack length. The results of the present model are shown as a solid line. The results of the numerical reference model is shown in gray. Dark gray for the continuum FEA and light gray for the beam-element FEA.

a renewed ERR rise. These findings underscore the pivotal role of slab mechanics in governing weak layer fracture processes and highlight boundary conditions as critical factors influencing ERR evolution.

Figure 6a shows that the effect of touchdown changes when more of the slab is resting on the collapsed weak layer. The slope of the center line changes significantly. At first touchdown the slab contacts the substratum at a higher angle and with further touchdown this angle reduces. This shows that a load transfer takes place and a smaller part of the weight load of the slab needs to be transferred through the weak layer, which limits the ERRs increase. This effect is nicely seen in Figure 6c. A local maximum of the ERR occurs at the first touchdown.

Then with further increase the ERR rises again until there is a steady state touchdown that results in a plateau value.

The progression through stages A, B, and C illustrates the intricate relationship between slab displacement, crack-tip loading, and energy release rate (ERR). The unsupported length, directly tied to the crack length in stages A and B, decreases in stage C as touchdown imposes a rotational constraint. This transition drives notable changes in ERR: an initial increase in stage A, a sharp drop at the onset of touchdown in stage B, and a steady state in stage C until boundary effects near the PST end cause a renewed ERR rise. These findings highlight boundary conditions as critical factors influencing ERR evolution.

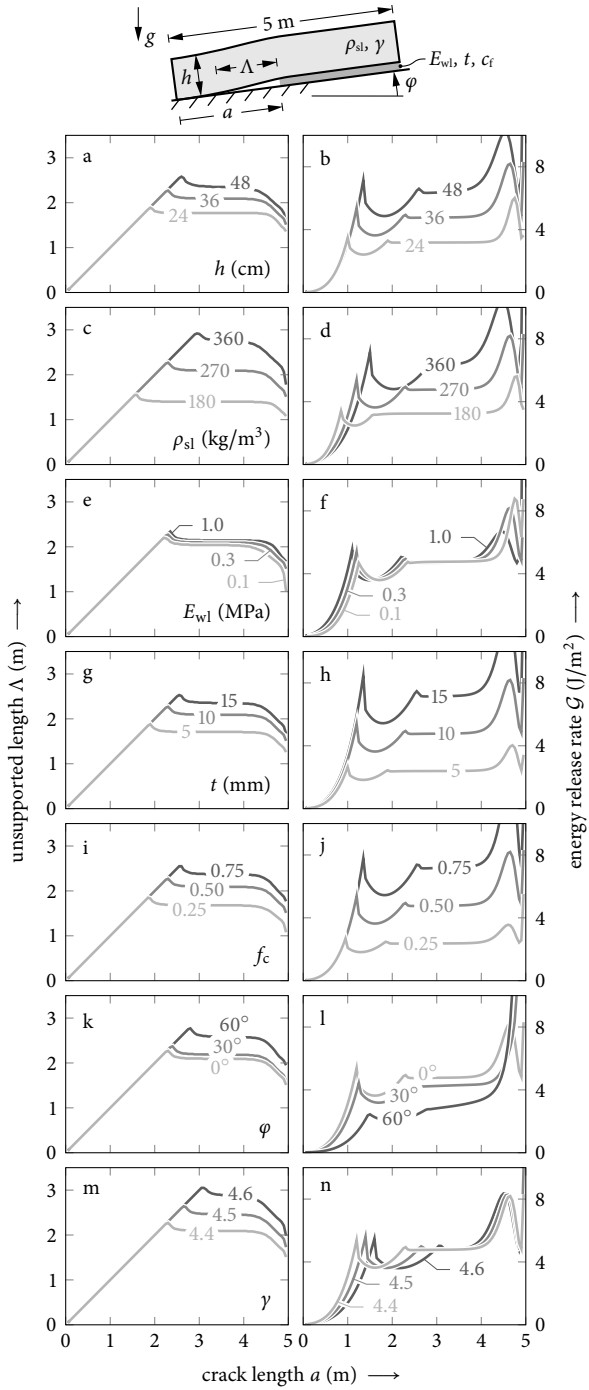

**Figure 7.** Influence of PST geometry and material properties on unsupported length $\Lambda$ (left column) and energy release rate $\mathcal{G}$ (right column): slab thickness $h$, density of the homogeneous slab $\rho_{sl}$, weak layer Young's modulus $E_{wl}$, thickness $t$, and collapse factor $f_c$, slope angle $\varphi$, and density scaling exponent $\gamma$, cf. Equation (12).

## 4.4 Influence of PST geometry and material properties

To understand the effect of parameters like the PST geometry and material properties, we conducted a sensitivity analyses for seven relevant parameters: slab thickness $h$, homogeneous slab density $\rho_{\mathrm{sl}}$, weak layer Young's modulus $E_{\mathrm{wl}}$, weak layer thickness $t$ and collapse factor $f_{\mathrm{c}}$, slope angle $\varphi$, and scaling exponent $\gamma$ for the elastic modulus parameterization in Equation (12). For each case the dependence of the unsupported length and the energy release rate on the crack length are shown in Figure 7.

Slab thickness governs the bending stiffness of the slab and leads to significant changes in the unsupported length and the energy release rate. We considered slab thickness (Figure 7a and b) values of $h = 24\,\mathrm{cm},\ 36\,\mathrm{cm}$ and $48\,\mathrm{cm}$. For thicker slabs the amplitude of the unsupported length increases and the transition of the stages (A to B to C) occurs at higher crack lengths. Similarly, the energy release rate also increases with the slab thicknesses. In stage A and stage B the energy release rate has the same initial increase but as the transition to stage C occurs at higher crack lengths, higher values of the energy release rate are obtained.

The effect of the density of the slab is shown in Figure 7c and d. Changing the density increases the loading of the PST and, by the density parametrization of the Young's modulus, the stiffness also increases. The unsupported length (Figure 7c) increases with increasing density as the effect of increased stiffness dominates the change of the bending behavior of the slab. The maximal unsupported length (i.e., the transition from stage B to C) occurs at longer crack lengths, when the density and hence the stiffness increases. The change from stage A to B can be seen in the corresponding ERR diagram (Figure 7d) as the first local maxima. While the corresponding crack length at which this transition occurs, increases only slightly, the magnitude of the ERR peak increases greatly with increasing density. For the density of $360\,\mathrm{kg/m^3}$ the plateau of constant ERR is very short, as the boundary effect of increasing ERR near the end of the PST affects a larger part of the PST. As increased density is linked with increased Young's modulus of the slab, not only the loading increases for higher densities but also the slab stiffness increases. Therefore, the effect of densities shares similarities with the pronounced effect of the slab thickness discussed above. Weißgraeber and Rosendahl (2023) discussed the effect of higher stiffness and increased weight loading of denser slabs by normalizing ERR for the average slab weight. They found that the two effects counteract each other, and mechanical modeling can help to understand which effect may dominate in a field experiment. This is of importance when new PST setups are developed, e.g. Birkeland et al. (2019); Adam et al. (2024).

In Figure 7e and f the effect of the weak layer Young's modulus $E_{\mathrm{wl}}$ is studied. When increasing this property the maximal unsupported length and the maximal energy release rate increases by a few percent. The maximum of the ERR occurs for slightly smaller crack lengths, when increasing the weak layer's Young's modulus. The weak layer thickness $t$ (Figure 7g and h) has an effect on the compliance of the weak layer and the domain of energy release during collapse. Increasing the weak layer thickness, increases both the unsupported length and the energy release rates. A very similar effect is observed when the collapse factor $f_{\mathrm{c}}$ is changed (Figure 7i and j).

Changing the slope angle $\varphi$ is analyzed in Figure 7k and l. For higher slope angles, the unsupported length increases, while the corresponding energy release rate decreases. At high slope angles, no clear plateau of the ERR is observed (Figure 7l). This can be attributed to the model not accounting for frictional sliding of touchdown segments, instead assuming full contact. The

observation that this effect only occurs at very high slope angles confirms that the chosen modeling approach, which does not

include frictional sliding, is sufficiently accurate for relevant slope angle regimes.

The scaling exponent $\gamma$ in the density-parametrization of the Young's modulus Equation (12) was varied between 4.4 to 4.6 to study the sensitivity on this parameter. An increase of the scaling exponent changes the transition from stage B to C (Figure 7m) and from stage A to B (Figure 7n) to higher crack lengths. This increases the magnitude of the maximal unsupported lengths but the magnitudes of the energy release rate does not change.

The sensitivity analysis reveals the key effects of PST geometry and material properties on unsupported length and energy release rate (ERR). Increasing slab thickness amplifies bending stiffness, delaying transitions between deformation stages and significantly raising ERR values. Higher slab density increases stiffness and weight loading, resulting in longer unsupported lengths, larger ERR peaks, and shorter ERR plateaus due to boundary effects. Weak layer properties, including higher Young's modulus, larger thickness, and larger collapse factors, consistently increase both unsupported length and ERR magnitude,

with the effects of thickness and collapse factor being particularly pronounced. Slope angle affects unsupported length and ERR inversely, with steeper angles leading to longer unsupported lengths but reduced ERR, and very high slopes disrupting ERR plateaus due to the absence of frictional sliding in the model. Lastly, increasing the scaling exponent for the density-parameterized Young's modulus shifts stage transitions to higher crack lengths, extending unsupported lengths but leaving ERR magnitude unchanged. These findings underscore the nuanced interplay between slab and weak layer properties in controlling

fracture processes.

## 4.5   Touchdown distance

In Figure 8 the corresponding touchdown lengths as function of slab and weak layer properties is addressed. To this end, both the unsupported length at first touchdown of the slab (transition from stage A to B) as well as the touchdown length associated with the steady state energy release rate in stage C are compared. Figure 8a shows the effect of the total slab thickness on the

touchdown lengths of a PST. Three different profiles are compared: a homogeneous slab, a stiff layered profile and a compliant layered profile (profiles H, C, D of Figure 3, respectively). A degressive increase of the touchdown length with increased slab thickness is observed for all profiles. The stiffest profile shows the highest values of the initial contact touchdown length and the steady state touchdown length. The touchdown lengths of the stiff profile C are about 70% higher than those of the profile D. The results for the homogeneous slab lie in between the profiles C and D. The touchdown lengths of profile C are about 30%

higher than those of the homogeneous slab. Both the effect of the layering and the effect of the total slab thickness highlight the great impact of the bending stiffness and layering on the touchdown length (Habermann et al., 2008; Weißgraeber and Rosendahl, 2023).

In Figure 8b the results of a parametric variation of the slab density of a homogeneous slab are shown for two different weak layer thicknesses. The change of the density has a twofold effect as both the gravitational loading increases linearly and the

stiffness of the slab increases with a power law due to the density parametrization Equation (12). The combined effect leads to an almost linear increase of the touchdown lengths with increased density of the slab. Increasing the weak layer thickness also significantly increases the touchdown lengths, as shown in Figure 8b for two different weak layer thickness values.

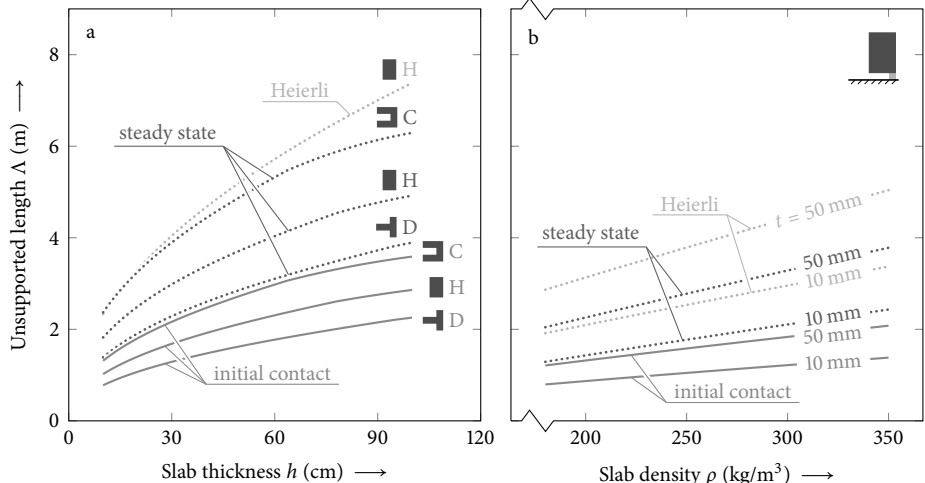

**Figure 8.** Analysis of the effect of slab and weak layer parameters on the unsupported lengths in propagation saw tests. a) Effect of slab thickness for different snow profiles. b) Effect of slab density and weak layer thickness. Both the unsupported length at first touchdown as well as the unsupported length of the steady state crack propagation in stage C are compared. The touchdown distance obtained using the model by Heierli (2005) as derived in Equation (14) is shown for comparison.

The parameter effects addressed in Figure 7 also show that the length associated with first touchdown and the unsupported length are strongly correlated, when governing parameters of slab or weak layer are changed. Within the parametric studies within this work we have observed that the ratio of the touchdown length of steady state conditions to the touchdown length of first touchdown shows only small variation with changes of parameters and typically lies between 1.6 to 1.8. This can be explained by the fact that both lengths are mainly dominated by the bending behavior of the slab-weak layer system.

We compare the results of the present model with the simplified model accounting for touchdown by Heierli (2005). Like our model, the Heierli model considered the stationary state and, thus, time-independence is assumed to simplify the differential equations. Heierli considers the transverse bending of thin plates using Euler-Bernoulli beam theory. Assuming the ends of the touchdown length to be slope-parallel ($w'(0) = w'(\Lambda) = 0$), one end is displaced by the amount of crack settlement ($w(\Lambda) = f_c t$), while the other end rests on the intact weak layer ($w(0) = 0$). The solution of this boundary value problem can be rearranged to yield the steady-state touchdown distance

$$\Lambda_{\mathrm{H}} = 2.331 \sqrt[4]{\frac{Eh^2 f_c t}{6\rho g \left(1 - \nu^2\right)}}, \tag{14}$$

where, in notation of the present model, $h$ is the total thickness of the slab, $\rho$ its average density, and $g = 9.81\,\mathrm{m/s^2}$ the earth's gravitational acceleration.

The touchdown lengths obtained with Equation (14) are shown in Figure 8a) and b) for comparison to the results of the present model. The touchdown lengths obtained from the Heierli model are always significantly longer than the touchdown lengths of the present model. Within the parameter studied here, a deviation by $34\,\%$ to $81\,\%$ is observed. The largest overesti-

mation of the touchdown length by the Heierli model was observed for thick slabs and thin weak layers. The overestimation of the touchdown lengths is rooted in the assumptions of the Heierli model which uses the Euler-Bernoulli beam theory that does not account for shear deformation in the slab. Yet shear deformation are especially significant for thick slabs and relatively short crack lengths. Heierli further assumes the unsupported slab to be clamped left and right and thus neglects the compliance of the rested and bedded slab, respectively. He further neglects the compliance of the weak layer itself. This leads to a zero rotation which subsequently induces a much longer touchdown distance. As the Heierli solution also does not account for the layering of the slab, further deviation occurs. For the considered cases the Heierli solution overestimates the touchdown lengths by a factor of three, when layered configurations are considered. Other aspects and especially those of the crack dynamics of the Heierli model are discussed in detail in Siron et al. (2023).

The touchdown length analysis underscores the significant influence of slab thickness, slab density, weak layer thickness, and layering on weak-layer fracture mechanics. Increasing slab thickness results in a degressive rise in touchdown lengths, with stiffer profiles exhibiting longer touchdown distances due to higher bending stiffness. Similarly, higher slab density leads to a near-linear increase in touchdown lengths, driven by combined effects of increased gravitational loading and stiffness. Weak layer thickness amplifies touchdown lengths substantially, highlighting its role in the compliance of the slab-weak layer system. The ratio of steady-state touchdown length to first touchdown length remains consistent, emphasizing the dominant role of bending behavior in these lengths.

### 4.6 Discussion of the model

In this section we discuss the model, its limitations and give an outlook on potential model improvements. The present model for slab touchdown is based on the mechanical model for layered slabs by Weißgraeber and Rosendahl (2023) and solves the nonlinear contact problem by introducing subsegements with linear behavior of initially unknown length, that are identified by finding the roots of polynomials of the touchdown length. The solution requires very low computational effort and allows for a solution within a few milliseconds on a standard desktop PC.

The present solution approach shares similarities with the approach used by Benedetti et al. (2019). They also proposed to use different stages (referred to as schemes) but the stages did not yield continuous transitions between the stages. Besides avoiding this limitation, the present model employs are more detailed representation of the slab allowing for shear, bending and extensional deformation by using the model of Weißgraeber and Rosendahl (2023), that also allows to consider effects of layering in the slab. Furthermore, the model allows to describe the conditions at the crack tip by providing the differential energy release rate. The model also accounts for the effect of the compliance of the weak layer, which has a strong effect on the accuracy of the stress and ERR results.

Validation has been performed by the comparison to the detailed FEM and DEM reference models. The comparison of the deflection for different crack lengths (Figure 4) shows that the very good agreement also covers the case of very long cracks in PSTs where a lifting deformation of the slab occurs. The results in Figure 6 for the energy release rate and for the unsupported crack length show a comparison to the non-linear FEA reference model. A very good agreement between the current model and the FE reference solutions is observed for both the unsupported crack length and the ERR. Both the deformation and

stress results in the study of the effect of the slope angle Figure 5 show generally very good agreement, but also a typical limitation of weak interface models. The stresses near the crack tip are not resolved, as discussed in Krenk (1992); Rosendahl and Weißgraeber (2020b).

In the present model, the stationary state of slab touchdown is considered, and dynamic effects are not accounted for. While this is directly applicable for describing the start of crack propagation after reaching the critical cut length in a PST experiment, it does not capture the dynamic processes that influence the ERR and stress fields once the crack transitions to a dynamic stage. During dynamic propagation, the ERR and stress distributions evolve rapidly, driven by the moving crack tip and its associated stress field Freund (1990). Although the observed ERR plateau remains conceptually valid, its magnitudes are expected to be influenced by these dynamic effects, which are beyond the scope of this work. A detailed discussion of dynamic processes and their impact, particularly in the context of the Heierli model, is provided by Siron et al. (2023). Importantly, when crack arrest occurs, the final arrested configuration represents a static state, which is well-described by the present model. However, the distinction between crack arrest that prevents dynamic propagation and crack arrest during dynamic propagation should be emphasized, as the latter involves processes not captured by this model. Future models should be extended to account for additional factors influencing crack arrest, including slab failure Gaume et al. (2015a), variability in weak layer failure properties, spatial variability Schweizer et al. (2008), and topography Gaume et al. (2019). In particular, spatial variability may play a crucial role in crack arrest, and further investigation,e.g. extended numerical analyses or uncertainty quantification, will be necessary to clarify the impact of spatial variability in further studies.

The present model assumes full contact and complete stress transfer during touchdown, without accounting for potential frictional movement along the weak layer. This assumption relies on the extensional stiffness of the slab being significantly higher than its flexural stiffness, such that the shear stresses generated in the weak layer during touchdown are assumed not to exceed the static friction threshold. Under this condition, a full transfer of shear stresses is possible, ensuring effective load redistribution. Additionally, the heterogeneous microstructure of the snow surface contributes to mechanical interlocking, which further facilitates shear stress transfer and enhances stability. However, this idealized representation may oversimplify the complex interactions occurring at the weak layer interface. In reality, localized frictional sliding or partial contact could alter the stress distribution, particularly in scenarios with variable weak layer properties or higher slope angles. These effects could influence the energy release rate and crack propagation behavior, highlighting the need for future research to integrate frictional effects into the model. Experimental studies and enhanced numerical approaches will be crucial to understanding how friction modulates slab touchdown and crack dynamics, thereby refining the predictive capabilities of the present model. Future improvements of this model could be aimed at taking the effect of friction into account.

### 4.7 Practical implications

We used the data reported by Bergfeld et al. (2023a,b) from up to $9\,\mathrm{m}$ long PSTs from January to March 2019 and identified data sets that allowed for unique identification of crack length and unsupported length. In this study they recorded the displacement of the slab using digital image correlation (DIC). From the obtained displacement fields the touchdown length can be retrieved and can be compared to the energy release rates obtained by use of the mechanical model from Weißgraeber and Rosendahl

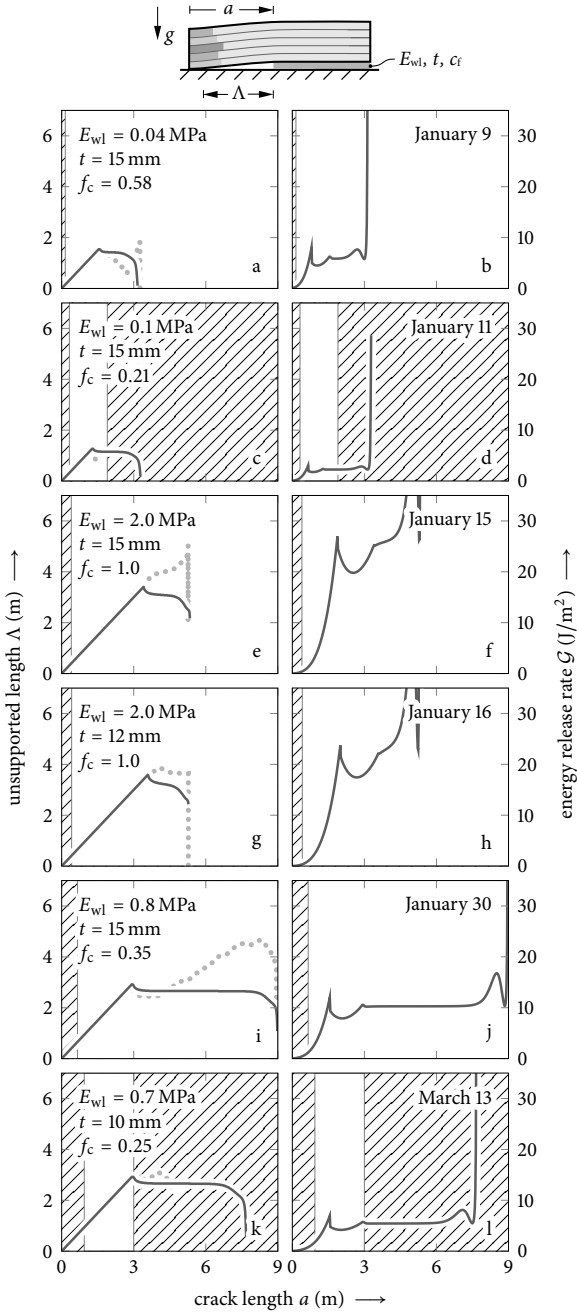

**Figure 9.** Crack propagation observed in PST experiments (Bergfeld et al., 2023a,b) of up to 9 m total length (for each length see Table 3). White background corresponds to unstable propagation while the hatched background indicates manual cutting and crack arrest. The properties of the weak layer, which vary from January to March 2019 (elastic modulus $E_{wl}$, thickness $t$, and collapse fraction $f_c$) were determined from a best fit of the measured (dotted) and modeled (solid lines) unsupported length $\Lambda$ (left column). Energy release rates $\mathcal{G}$ at corresponding crack lengths $a$ are given in the right column.

**Table 3.** Critical ($\mathcal{G}_\mathrm{c}$) and steady-state ($\mathcal{G}_\mathrm{s}$) energy release rates (ERR) in the PST experiments of 2019. Bergfeld et al. (2023b,a)

| Date | PST result | Fracture toughness $\mathcal{G}_\mathrm{c}$ (J/m$^2$) | Steady-state ERR $\mathcal{G}_\mathrm{s}$ (J/m$^2$) | Ratio $\mathcal{G}_\mathrm{c}/\mathcal{G}_\mathrm{s}$ (–) | PST length (m) |
|------|------------|------|------|------|------|
| Jan 9 | END | 0.360 | 5.9 | 0.061 | 3.2 |
| Jan 11 | ARR | 0.592 | 2.3 | 0.257 | 3.3 |
| Jan 15 | END | 0.391 | 25.0 | 0.016 | 5.4 |
| Jan 16 | END | 0.353 | 22.0 | 0.016 | 5.3 |
| Jan 30 | END | 1.212 | 10.3 | 0.118 | 9.0 |
| Mar 13 | ARR | 1.770 | 5.4 | 0.328 | 7.7 |

(2023) (details are given in Bergfeld et al. (2023a)). A best fit of the measured and modeled unsupported length was done to account for possible temporal variations of the weak layer properties during the course of the season as well as spatial variations within the field test site. The obtained values are shown in Figure 9 along with the curves.

For six long PST experiments the observed unsupported lengths and the ERR are shown in Figure 9. In all experiments the onset of unstable crack propagation is observed at critical crack lengths that are still in stage A according to the present model. This result is consistent with Bergfeld et al. (2023a) where no such touchdown is reported. The corresponding critical energy release rates take values of $0.43\,\mathrm{J/m}^2$ to $2.93\,\mathrm{J/m}^2$.

The model results for the considered experimental configurations show the characteristic curves of unsupported length with its maximum at the transition from stage B to C and of the ERR with the local maximum at first contact and the plateau for steady state. The values of the critical ERR (fracture toughness) that are obtained for the critical crack length are given in Table 3 along with the ERR during the steady state crack propagation. The ratio of these ERR values lies between $0.016$ to $0.118$ for the PST experiments which resulted in full propagation throughout the beam (END). For the experiments on January 11 and March 13 where crack arrest (ARR) was observed, the ratio takes values of $0.257$ and $0.328$. This ratio may be a decisive factor to understand the behavior of dynamic crack propagation and conditions of crack arrest in dry-snow slab avalanches. However, only the PST experiments of January 30 and March 13 are long enough ($9.0\,\mathrm{m}$ and $7.7\,\mathrm{m}$, respectively) to exhibit a distinct and extended plateau of the ERR in stage C.

For the experiments with crack arrest, the model predicts a touchdown of the slab (transition from stage B to C). In the crack arrest case of the Jan 11 experiment, the crack arrest occurred at around 60cm after the touchdown. In the other experiment with crack arrest (Mar 13), the arrest length coincides well with the modelled touchdown length.

The touchdown has a strong effect on the ERR in PSTs. The decrease in energy release rate after the transition from A to B is assumed to be one of the primary factors driving crack arrest. As expected the model also shows that there is plateau value of the ERR, when full touchdown has developed and no edge effects come into play (yet). The ERR value of this plateau is definitely the most relevant for propagation of weak layer failure over a long distance. The results indicate that the PST

experiments must be done with very long PSTs to obtain a distinct plateau. This finding coincides with the findings of Bergfeld et al. (2023a), as they report that PSTs must be longer than one touchdown distance to study whether a crack can propagate a long distance. Near the end of the PST the ERR rises again. Also because of this effect PSTs are required to be sufficiently long to obtain reliable results (Van Herwijnen et al., 2010; van Herwijnen and Birkeland, 2014). Bair et al. (2014) studied this
at hand of PSTs of different length and also found that for steady state very long PSTs must be studied. In their study they observed that with increasing length of the PSTs less full propagation results occurred.

The ability of a weak layer crack to propagate over a large area is likely determined by the steady-state ERR in C and its ability to pass through the local minimum of the ERR in B (Rosendahl et al., 2019b; Bergfeld et al., 2023b). Another possible factor limiting crack propagation within a slope could be slab fracture that limits the bending deformation (Schweizer et al.,
2014b; Gaume et al., 2015a).

The practical implications of this study highlight the critical role of slab and weak layer properties, as well as PST geometry, in determining weak layer crack propagation and arrest. The model demonstrates that the energy release rate (ERR) is highly sensitive to slab thickness, density, and weak layer properties, with the steady-state ERR in stage C being the most relevant for long-distance crack propagation. The findings emphasize the necessity of conducting very long PST experiments to capture a
495 distinct ERR plateau and accurately assess the potential for sustained crack propagation. Touchdown, marked by the transition from stage B to C, significantly reduces the ERR and is identified as a primary factor in crack arrest, aligning well with field observations. The study also reveals that slab fracture and dynamic effects, although not explicitly modeled, may further influence crack arrest and propagation in real-world scenarios. These results offer insights for designing effective PST experiments and developing best-practice guidelines for assessing snowpack stability and avalanche risk.

## 5 Conclusion

We have provided an efficient solution approach for the non-linear problem of slab touchdown during propagation saw test experiments, that are used to characterize fracture mechanical properties of weak layers. The solution is highly efficient and shows very good agreement with numerical reference models (Finite Element and Discrete Element Methods). In addition experimentally measured touchdown lengths from up to $9\,\mathrm{m}$ long PSTs are reproduced well. In its current implementation, the
505 model does not account for friction or dynamic effects.

Across the provided parameter study, all modeled slab-weak layer configurations have consistently led to i) a local maximum of the ERR when slab comes into contact with the substrate (transition from stage A to B) and ii) a maximum of the touchdown length at the transition from B to C closely before the steady state touchdown length reached in stage C. At this stage the touchdown length is not coupled to crack length anymore and the mechanical deformation of the slab-weak layer system is
510 translationally invariant with respect to the crack tip. Hence, in stage C also the ERR reaches a steady state value which is lower than the local maximum at transition from A to B. The behavior of the ERR with crack length has practical implications. PST experiments can result in crack arrest after a couple of meters Bergfeld et al. (2023b). After the local maximum the reduced ERR in the steady state (stage C) provides a reasonable explanation for crack arrest phenomena after initially unstable

cracking. This finding also suggests that large-scale crack propagation, a necessity of avalanches, is more likely for a snowpack with high steady state ERR. If the ERR at the onset of crack propagation, namely the fracture toughness $\mathcal{G}_c$ of the weak layer, is close or even higher than the steady state ERR toughness $\mathcal{G}_s$, this suggests that large-scale crack propagation is more unlikely. As shown in Table 3, the quotient of both $\mathcal{G}_c/\mathcal{G}_s$ is therefore a reasonable metric to map the propensity for wide-spread crack propagation of a given snowpack/PST.

Overall, the work provides important insights into the physical processes during onset of crack propagation and potential crack arrest in PST experiments.

## Appendix A:  Additional comparisons with previous models

Benedetti et al. (2019) adopt a modeling approach similar to that of the present work. The key distinction lies in the present model's explicit consideration of weak-layer compliance. The impact of this difference is illustrated in figure A1. For homogeneous slabs with a rigid weak layer, Benedetti et al. (2019) derive the following unsupported lengths at the transitions between stages A and B (initial contact),

$$\Lambda_{A \to B} = \sqrt[4]{\frac{2 E_{sl} h^3 t}{3 q_n}},\tag{A1}$$

and between stages B and C (full contact),

$$\Lambda_{B \to C} = \sqrt{3}\,\Lambda_{A \to B}.\tag{A2}$$

The assumption of a rigid weak layer in these equations results in significantly larger unsupported lengths compared to both the present model and the finite-element reference model (figure A1). Additional details in the model assumptions are listed in table A1.

*Code availability.*  The present model is made publicly available as Python code under https://github.com/2phi/weac.

*Author contributions.*  PW and PLR conceived the study. JS, PW, and PLR developed the theoretical framework. GB, BB, and AH collected the field data. JS, FR and GB developed the reference models. All authors contributed to the interpretation of the results and writing of the manuscript.

*Competing interests.*  The authors declare that they have no conflict of interest.

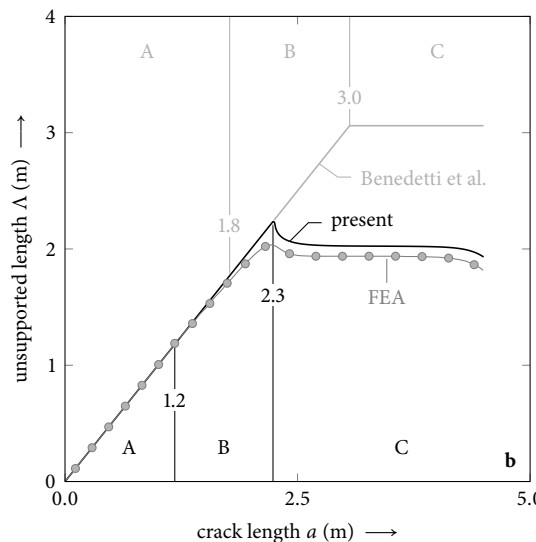

**Figure A1.** Comparison of unsupported length and stages between Benedetti et al. (2019) (gray), the present work (black), and the FEA reference solution (circles). Material properties and dimensions are given in table 2.

*Acknowledgements.* This work was funded by Deutsche Forschungsgemeinschaft (DFG, German Research Foundation) under grant no. 460195514 and the Swiss National Science Foundation (grant no. 200021_169424 and 200021L_201071). We used deepl.com/write to improve ease of reading and comprehension. We acknowledge support by the German Research Foundation and the Open Access Publishing Fund of Technische Universität Darmstadt.

**Table A1.** Comparison between Benedetti et al. (2019) and the present model

|  | **Benedetti et al.** | **Present Model** |
|---|---|---|
| **Slab kinematics** | Homogeneous Euler–Bernoulli beam theory | First-order shear deformation theory to account for short beams |
| **Slab layering** | Not accounted for | Laminated plate theory, capturing bending–extension coupling and the influence of slab layering on bending and extensional stiffnesses |
| **Weak layer compliance** | Not accounted for | Winkler foundation |
| **Stress analysis** | Rigid beam segment assumption, leading to linear stress distribution along the length | Weak interface solution of shear and normal stresses based on the set of differential equations of the slab deformation (bending deformation, shear deformation, and axial extension) |
| **Stages** | | |
| **Stage I / Stage A** | Bending of slab | Bending of slab and weak layer deformation, energy release rate of weak-layer crack propagation |
| **Stage II / Stage B** | No moment introduced at contact point, initial-contact length is a function of (isotropic) slab properties | No moment introduced at contact point, initial-contact length is a function of layered slab and weak-layer properties |
| **Stage III / Stage C** | Slab face is vertical at the contact point (transition from II to III is discontinuous), full-contact length is a function of (isotropic) slab properties | Moments introduced by the slab resting on the collapsed weak layer are represented by rotational springs, whose stiffness is obtained from a full slab deformation analysis by the principle of virtual forces, full-contact length is a function of layered slab and weak-layer properties, energy release rate of weak-layer crack propagation |

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
