# Peer review of "The effect of slab touchdown on anticrack arrest in propagation saw tests"

_Natural Hazards and Earth System Sciences, 2024_

## Referee Comment (RC1)

[referee-annotated manuscript omitted]

---

## Author Comment (AC1)

**Revision of: *The effect of slab touchdown on anticrack arrest in propagation saw tests**

Dear Editors and Reviewers,

Thank you for your thorough review and constructive feedback on our manuscript. We appreciate the opportunity to revise our work based on your comments and suggestions. We carefully considered each point and made corresponding revisions to enhance the clarity, accuracy, and impact of our manuscript.

In this document, we address each remark (*blue, italic*) point-by-point (black). We hope that these modifications adequately address the concerns raised and improve the manuscript. We are grateful for the guidance that the reviewers' expertise has provided and are confident that these changes have strengthened our submission.

**Referee #1**

*The authors present a novel method for analyzing slab touchdown in a Propagation Saw Test (PST) by developing a closed-form analytical model that incorporates mixed-mode loading (compression and shear) to calculate the Energy Release Rate (ERR) from slab deformation and the resulting crack arrest. To streamline the mathematical framework, they divide the slab touchdown progression into three distinct stages, each characterized by linear behavior and representing different phases of crack propagation and interaction with the collapsed weak layer. The model is validated using finite-element and discrete-element methods, providing a more nuanced understanding of avalanche initiation mechanisms.*

*I enjoyed reading this article. It is well-written, the model is clearly explained, and sensitivity analyses clearly illustrate its contribution to the avalanche field. While I have very few specific comments on the article's content, minor additions could enhance the manuscript. Below is a list of general comments.*

**Remark 1.1** *Sensitivity Analysis: The article provides a thorough sensitivity analysis examining the impact of various parameters—such as slab thickness, density, weak layer properties, and slope angle—on crack propagation and arrest. Including a summarized list or table of these findings would enhance their accessibility and value to a broader audience.*

We appreciate the reviewer's suggestion and agree that clarity is important. However, summarizing the findings in a table might oversimplify the complex interdependencies of parameters such as slab thickness, density, weak layer properties, and slope angle. To address this, we will revise the manuscript to include clearer summaries within each relevant section, ensuring that the results remain accessible without losing context.

**Remark 1.2** *Model Limitations in Capturing Dynamic Processes: Although the study's static model effectively illustrates the effects of slab touchdown, it fails to capture the dynamic processes involved in slope-scale crack propagation during real-world avalanches. While the authors acknowledge that the model does not account for dynamic stages of crack propagation, this limitation should be explicitly emphasized in the discussion or conclusions. In practical terms, crack arrest in real-world avalanches, occurring after the weak layer crack transitions to a dynamic stage, differs significantly from crack arrest that prevents the transition to a dynamic stage. Addressing this distinction is crucial for clarifying the model's applicability and limitations.*

We agree that the model's limitation in capturing dynamic crack propagation in real avalanches is important. We will emphasize this distinction more clearly in the revised discussion and conclusion, highlighting the difference between crack arrest before and during the dynamic stage.

**Remark 1.3** *Limited Experimental Validation: While the authors validate the model using numerical simulations, there is limited reliance on direct experimental data to substantiate the model's predictions. The use of experimental data from natural winter snowpack (Bergfeld et al., 2023a, b) is only briefly mentioned in the "Practical Implications" section. To strengthen the manuscript, I recommend relocating the sections that compare the model to data from Bergfeld et al., 2023a, b to the "Methods" and "Results" sections, as this comparison provides the most compelling validation of the proposed model.*

While we provide some discussion of experimental comparisons in Section 4.7, we recognize the importance of better aligning these findings with the broader narrative of the manuscript. We will improve the integration of these comparisons with the methodological framework and ensure they are more clearly referenced earlier in the text. This will help highlight their relevance to the model's validation while maintaining the logical flow of the manuscript.

**Remark 1.4** *Complexity and Accessibility: The article's technical language and in-depth mathematical modeling may limit its accessibility to a broader audience, such as those involved in developing "best practice" guidelines for snowpack observation and avalanche testing who could benefit from its insights. To increase its impact, the authors could provide a concise summary of the key results and a more accessible overview, highlighting practical implications and recommendations for application in the field.*

We will revise the conclusion to include a concise summary of key results and practical implications for broader accessibility. Additionally, we aim to seek cooperation with field experts to apply such models in understanding real-world implications. We are looking forward to any exchange here.

**Remark 1.5** *Practical Implications Not Fully Explored: While the study offers theoretical insights into crack arrest mechanisms, its practical implications are mostly confined to tentative recommendations for PST geometry. However, as the authors suggest by listing these recommendations, even the optimal geometry for PST remains uncertain, and there is still a lack of understanding of how to interpret test results concerning actual avalanche occurrences. The article would benefit from leveraging its rigorous sensitivity analysis to discuss further how these findings could inform practical avalanche safety strategies and improve real-world decision-making for avalanche risk management.*

We acknowledge the importance of connecting the theoretical insights presented in this study to practical applications in avalanche risk management. In the revised manuscript, we will expand the discussion to address the practical implications of our findings. Specifically, we will highlight how the sensitivity analysis can inform optimal PST geometry and its interpretation. Additionally, we will discuss the limitations and uncertainties associated with translating test results to real-world avalanche occurrences and suggest avenues for improving decision-making strategies in avalanche safety.

**Remark 1.6** *Simplification and Assumptions: As noted in the article, the model does not consider frictional sliding during slab touchdown and assumes full contact and stress transfer. In real-world scenarios, frictional effects can be significant, particularly at high slope angles, potentially affecting the ERR and the probability of crack arrest. Although this limitation comes into play at the extreme upper end of the avalanche release slope angle, this and other limitations should be clearly outlined in the discussion or conclusion sections to provide a more comprehensive understanding of the model's constraints and applicability to real avalanche conditions.*

The role of friction during initial contact requires detailed exploration, which we plan to address in future research alongside experimental analyses. We will clarify this limitation in the revised discussion and conclusion.

**Remark 1.7** *Detailed hints and remarks in the attached annotated PDF file of the manuscript.*

Thank you for the direct annotations in the manuscript. They are all helpful to increase clarity of the points we want to bring across. We will consider all in the revised version of the manuscript.

---

## Author Comment (AC2)

**Revision of: *The effect of slab touchdown on anticrack arrest in propagation saw tests**

Dear Editors and Reviewers,

Thank you for your thorough review and constructive feedback on our manuscript. We appreciate the opportunity to revise our work based on your comments and suggestions. We carefully considered each point and made corresponding revisions to enhance the clarity, accuracy, and impact of our manuscript.

In this document, we address each remark (*blue, italic*) point-by-point (black). We hope that these modifications adequately address the concerns raised and improve the manuscript. We are grateful for the guidance that the reviewers' expertise has provided and are confident that these changes have strengthened our submission.

**Referee #2**

*The paper is well-written and presents solid research, though its contribution is somewhat incremental compared to existing literature, particularly with respect to previous work by both the authors and other scholars in the field. Despite this, I believe the study introduces enough novel insights to warrant publication in NHESS after a few major revisions detailed below together with more specific minor comments.*

**Remark 2.1** *Novelty and contribution: As briefly mentioned above, although the paper builds upon previous work by the authors and others in the field (especially Benedetti et al.), it does introduce enough novel elements to warrant consideration for publication in NHESS. A significant portion of the mechanical model has already been presented in Weissgraber and Rosendahl (2023) and applied in Bergfeld et al. (2023). Moreover, the concept of touch-down distance in a static slab-weak layer model was introduced by Benedetti et al., although without incorporating slab shear (Euler-Bernoulli beam theory) nor slab layering. The important contribution of this paper lies in combining these two elements, leading to a more robust model, performing detailed sensitivity analysis, validation with FEM and DEM, and applying it to field data. I believe this combination is very solid and represents excellent work. However, I encourage the authors to be more explicit about the overlap with prior work and to clearly articulate the unique contributions of this paper.*

The model is based on the well-validated and established WEAC model. However, it employs a completely new set of modeling techniques to allow for the consideration of slab contact, which was not possible in WEAC. To this end, we extend WEAC by incorporating slab contact.

We appreciate your suggestion to outline the differences from Benedetti's model more clearly. We will also acknowledge that Benedetti et al. introduced the first mechanical model for this aspect. Our model incorporates several aspects not covered by Benedetti et al., which are known to be critical for the meaningfulness of mechanical analyses:

- We use a first-order shear deformation theory because the beam sections are often too short for the normality assumption of Euler-Bernoulli beam theory. Additionally, stress concentrations in the weak layer lead to localized line loads along the beam length, violating this assumption.

- Slab layering significantly affects stiffness (bending and extensional) and generally leads to bending-extension coupling. To account for this, we use laminated plate theory to represent slab behavior accurately.

- The compliance of the weak layer substantially alters the slab-weak-layer system, influencing energy-driven failure processes. To address this, we use the WEAC model, validated with finite element analyses based on weak-interface modeling and Winkler foundations for weak-layer compliance.

- We use the weak interface model to identify the shear and normal stresses in the weak layer. Here the Benedetti et al work uses a very strong simplification of a rigid beam segment that shows a rigid body movement at the center of gravity of the unsupported slab. This leads to unrealistic, linear stress distributions along the weak layer interface.

- Current research has shown that PSTs can be used to identify the fracture energy of a shear crack or anticrack within the weak layer and the concepts of fracture mechanics apply. To study this brittle fracture phenomena, we use the energy release rate at the crack tip (stress intensity factors could also be used, but are not well defined for weak interface models). We compute the energy release rate by using the concepts of weak interfaces as this is also established for the fracture mechanics analysis of adhesive joints. In a full continuum analysis the crack tip would show a stress singularity within the linear-elastic analysis, which would lead to vanishing critical loads (the smallest loading leads to infinite stresses) and hence we see, that a strength-based criterion cannot applied to such strong stress concentrations. Weak interface models are not able to cover the stress singularity but still provide meaningful results with strong local stress concentrations at the end of the intact weak interface but weak interface model still allow for the validated computation of differential energy release rates.

- To render the evolving contact of the slab on the substratum we use a separation into stages like Benedetti et al did. The major improvement is that we have used a set of springs to account for the interaction of the unsupported slab with the slab with intact weak interface and with the part of the slab already in contact with the substratum. The spring stiffnesses are obtained by using a full beam model with the above-described features as it is implemented in weac. This represents an important expansion of the weac model.

**Remark 2.2** *Some overstatements and comparison/discussion with Benedetti: I noted a few overstatements throughout the paper that I believe are unnecessary. Additionally, it would be good to add a comparison with Benedetti's "twin" model or at least a deeper discussion. For instance, the paper introduces the different phases during collapse, which were already presented in greater detail (with additional steps) by Benedetti et al. It appears from the methods section that the authors are positioning themselves as the first to introduce these phases, but they were previously discussed by Heierli (though in less detail), Benedetti et al., and more recently by Siron et al., who also included dynamics. While this is briefly acknowledged in the discussion, it should be much clearer earlier in the paper.*

We will carefully review the manuscript for potential overstatements and make adjustments where necessary. Additionally, we will explicitly acknowledge and clearly reference the contributions of Heierli, Benedetti et al., and Siron et al. regarding the concept of stages during collapse. While we intended to build upon these prior works, we understand the importance of highlighting their foundational contributions earlier in the manuscript. In the revised document, we will ensure these aspects are presented more transparently and thoroughly.

**Remark 2.3** *Model comparison: I would suggest comparing the current model with Benedetti's analytical model under various configurations (e.g., without layering), particularly regarding unsupported length and touch-down distance. This would better illustrate the necessity of including slab shear. For example, slab stress in Benedetti has characteristics similar to the ERR in your paper, but this trend is not reflected in Benedetti's weak layer stresses. Elaborating on the differences would be great: is the discrepancy due to incorrect assumptions (e.g., lack of slab shear) or something else? The claim that Benedetti's model is not continuous also needs clarification. The issue seems to be with derivatives, which are often discontinuous in contact and friction problems. A more precise statement would help.*

Comparing both approaches shows that Benedetti's model can be considered a special case of the present model, particularly in the first stage, due to similar boundary conditions. Equation (8) of the in our manuscriupt

$$0 = \frac{a_{\mathrm{AB}}^4}{8K_0} + \frac{a_{\mathrm{AB}}^3}{2k_{\mathrm{R}}^\circ} + \frac{a_{\mathrm{AB}}^2}{2\kappa A_{55}} + \frac{a_{\mathrm{AB}}}{k_{\mathrm{N}}^\circ} - \frac{f_c t}{q}, \tag{1}$$

reduces to Benedetti's solution

$$L_{\mathrm{IC}} = \sqrt[4]{\frac{8E_s I_s h_w}{q_v}} \tag{2}$$

under the following assumptions:

- The compliance of the weak layer on which the supported section of the slabs rests is assumed to be infinitely high. No deformation within that section of the weak layer. Both the normal stiffness relevant for the transverse section force in the slab $k_{\mathrm{N}}^\circ$ as well as the rotational stiffness $k_{\mathrm{R}}^\circ$ relevant for the bending moment in the beam are set to infinity.

- The shear stiffness of the slab without foundation $\kappa A_{55}$ is set to infinity as well, allowing for no shear deformation within the slab.

- For the effective (cylindrical) bending stiffness of the slab $K_0 = D_{11} - B_{11}^2 A_{11}^{-1}$ is obtained under consideration of no bending-extension coupling in the slab (e.g. by a layering which is not symmetric to the mid-line of the slab)

- Further the bending stiffness $D_{11}$ no effect of layering is assumed and a homogeneous, isotropic material is assumed within the slab.

- It assumed that the collapse height equals the thickness of the weak layer.

With these assumptions eq. 8 reduces to the polynomial of $L^4$ only and the solution by Benedetti et al. is obtained for stage I. The largest difference between the two approaches lies in the weak-layer compliance.

We will make it clearer in the revised document that equation (8) includes the special case of the Benedetti et al. solution.

**Remark 2.4** *Crack arrest and model relevance: The link to crack arrest is only introduced late in the discussion, which is somewhat confusing. In addition, while I appreciate the use of fracture mechanics, the paper introduces the critical energy release rate ($G_c$), essentially stress ($a^2$) divided by weak layer stiffness, adding another layer of uncertainty as one needs to know $k_n$ and $k_t$ as well, properties which are very hard to measure. I recommend instead, or at least additionally, to present results in a stress-strength framework to avoid this additional uncertainty and offer further insights into crack arrest. For instance, including metrics like $\tau_c/\tau_s$ and $\sigma_c/\sigma_s$, alongside $G_c/G_s$, would be beneficial, as these are commonly used in engineering and avalanche science to assess stability and also in many strength-of-material oriented models. This would be a very interesting additional outcome that could give insights for researchers using different frameworks (fracture or strength of materials). In fact, the sentence "As shown in Table 3, the quotient of both $G_c/G_s$ is therefore a reasonable metric to map the propensity for widespread crack propagation of a given snowpack/PST" suggests that one can only get this information through toughness. You would essentially get the same type of information by looking at stress and strength. Additionally, this would make a better link to Figure 5 (DEM comparison), which is the only one with stress instead of energy... I think this would be a very valuable and interesting addition.*

Thank you for bringing up this important point. It touches upon a common misunderstanding concerning stress behavior at the tips of sharp notches (e.g., V-notches, crack tips, anti-crack tips, etc.). According to elastic continuum mechanics, stresses at these locations are theoretically infinite. However, many models yield finite stress values because they lack the necessary resolution to accurately capture the stress singularity at the crack tip.

In weak-interface models like ours, the weak layer of finite thickness is considered as a whole, and the crack itself is not explicitly resolved. In finite-element analyses, the stress at the notch tip increases as the element size decreases around the notch tip, regardless of how small the elements are chosen. This is due to the polynomial basis functions used in these elements. Similarly, discrete-element models employ elements of finite size whose geometries differ significantly from the microstructure of a weak layer. Inter-particle interactions are modeled using spring-type connections between the particles' centers of mass, and decreasing the particle size leads to increased observed stresses.

Consequently, the stresses observed at crack tips are influenced by the chosen characteristic length scale (element size, particle size, etc.). While the stress solutions provided by these models are generally correct, they are not accurate in the immediate vicinity of a notch. Therefore, it is not physically appropriate to use such stress values for stability assessments.

To illustrate this, consider two weak layers with identical elastic properties (Young's modulus, Poisson's ratio) and the same cut length but different thicknesses. Although the crack-tip stresses in the thicker weak layer are smaller, its energy release rate is higher, making it the weaker layer! This phenomenon has been demonstrated both theoretically and experimentally, for instance, in studies of adhesive layers, which share similar geometric and fracture characteristics with weak snow layers in terms of elastic contrast and failure localization Stein et al. (2015).

**Remark 2.5** *Another important aspect is the lack of discussion regarding the model's relevance to real-scale avalanches. The paper focuses on arrest conditions at the PST scale, not the slope scale. Although arrest can indeed be induced by slab and weak layer properties at the PST scale, as demonstrated here and by others, once crack propagation occurs at the slope scale, touch-down distance does not have the same impact (it actually becomes much larger in a dynamic setting). Arrest at this stage is likely driven by slab tensile failure, spatial variability (see Meloche et al. https://arxiv.org/abs/2406.01360), and topography (Gaume et al. 2019 Cold Reg.). This scale is reflected very well in the title of the paper, but the text needs further elaboration in the introduction and discussion sections in my opinion.*

We have revised the manuscript to more clearly define the scope of this study in both the Introduction and Discussion sections.

**Remark 2.6** *Crack arrest explanation: Moreover, the current paper does not completely explain why crack arrest occurs; it suggests that it could be related to $G_c/G_s$, but the ratios remain below one, meaning energy release still exceeds toughness. This is fine, but I think it should be further discussed. Additional research is needed to fully understand this process, especially at the slope scale, where other factors like slab tensile failure, spatial variability, and topography play crucial roles.*

We completely agree with this observation. While we aimed to uncover a conclusive explanation for why crack arrest occurs, our findings only provided preliminary insights. This underscores the need for further studies, which we have now emphasized in the revised Discussion section.

**Remark 2.7** *Frictional sliding: It is both surprising and a bit disappointing that the current study does not account for frictional sliding, an essential factor, particularly for cases on steep slopes. Why was this omitted? Including friction seems quite straightforward and would provide a more comprehensive understanding of the crack arrest phenomena. I understand that at a scale of a classical PST (1m) it won't affect the results much, but at the larger scale, it will have crucial consequences on the results. There must be a clear and objective rationale for excluding it, though I find it difficult to grasp the complexity behind this decision, especially since similar modeling has included friction before. How complex is it to add the friction term into the equations?*

Incorporating friction into the model introduces physical nonlinearity, necessitating an iterative solution scheme. While this is beyond the scope of the current study, we plan to include it in a future iteration of the model.

**Remark 2.8** *Equations: I suggest adding more details on the stress calculations.*

The limited emphasis on stresses was a deliberate choice. As explained in our response to R2.4, stresses are not directly relevant in the context of crack propagation problems, such as those studied in propagation saw tests.

**Remark 2.9** *Experimental validation: At the moment, the validation is performed using DEM and FEM, and the model is then applied to experimental cases. I wonder if it would be feasible to extend the validation to a much broader database of PSTs, which include snowpack information and PST outcomes, even if this validation is at a lower level of detail. One does not always have such precise data as those presented here, but I still think such data could give important general insights into the model applicability. For instance, one could use a threshold-based approach to check whether you are able to "predict" ARR and END cases.*

Based on your suggestion, we analyzed all PSTs with complete snow profiles available in the SnowPilot database ( 2500 data points). However, we identified a significant bias in PST datasets: nearly all PSTs in the database are 1 m long, which means crack arrest is only observed for PSTs with very short cut lengths, i.e., when the weak-layer fracture toughness is exceptionally low. This introduces an unrealistic bias, and a substantial number of longer PSTs would be required to draw reliable conclusions from the proposed analysis.

**Remark 2.10** *Suggested limitations and outlook section: A section on limitations and outlook would greatly enhance the paper. It could include topics such as the effects of slab fracture (which could be modeled similarly to Benedetti but here including layering model) and discuss skier triggering, spatial variability, and slab fracture dynamics, especially since slab fracture may be difficult to observe in PSTs.*

Slab fracture can be calculated using the stress solutions provided in Weißgraeber and Rosendahl (2023). However, this requires careful consideration, as slab fractures can initiate either from the top or the bottom of the slab.

Skier triggering is a nucleation phenomenon governed simultaneously by stress and toughness, necessitating a dedicated modeling approach. In this context, slab touchdown plays a significant role in large-scale crack propagation but is less critical during the initiation phase. To address this in the future, we are developing an experimental setup to validate skier-triggering models against empirical observations. However, this effort is beyond the scope of the current paper.

The characteristics of spatial variability discussed in prior publications also apply here. Unfortunately, we do not have touch-down PST data available to provide additional insights into this aspect.

**References**

Stein, N., Weißgraeber, P., and Becker, W.: A model for brittle failure in adhesive lap joints of arbitrary joint configuration, Composite Structures, 133, 707–718, 2015.

Weißgraeber, P. and Rosendahl, P. L.: A closed-form model for layered snow slabs, The Cryosphere, 17, 1475–1496, https://doi.org/10.5194/tc-17-1475-2023, https://tc.copernicus.org/articles/17/1475/2023/, 2023.

175

---

## Author Response (AR2)

**Revision R2 of: *The effect of slab touchdown on anticrack arrest in propagation saw tests**

Dear Anonymous Referee 2 Thank you for your thorough review and constructive feedback on our manuscript. In this document, we address each remark (*blue, italic*) point-by-point (black). We hope that these modifications adequately address the concerns raised and improve the manuscript. We are grateful for the guidance that the reviewers' expertise has provided and are confident that these changes have strengthened our submission.

**Referee #2**

*In general, the authors have done a decent job of addressing my questions and comments. However, I feel that a few important points were overlooked, and I would like to bring them to their attention. Once these are addressed, I am confident that the paper will be of high quality and suitable for publication in NHESS.*

**Remark 2.1** *Comparison with Benedetti: While a brief discussion of the developed model with that of Benedetti is provided in Section 2.2, no quantitative comparison is offered. The text states that stage A and the transition from A to B are similar to Benedetti, but no further comparison is made beyond this observation. This is surprising, as major revisions were requested and the effort required to make such a comparison seems minimal. I would appreciate it if a more detailed comparison in different stages (graphical, e.g. in case similar to Fig. 6) could be included, ideally in the main paper, or at least in a supplement/appendix or peer-reviewed document. This would further highlight the advantages of their model compared to Benedetti's.*

In the revised document at the beginning of section 2, we have detailed the differences of the modeling approach. To expand on this, we have added the following section to the appendix:

**A   Additional comparisons with previous models**

Benedetti et al. (2019) adopt a modeling approach similar to that of the present work. The key distinction lies in the present model's explicit consideration of weak-layer compliance. The impact of this difference is illustrated in Fig. A1. For homogeneous slabs with a rigid weak layer, Benedetti et al. (2019) derive the following unsupported lengths at the transitions between stages A and B (initial contact),

$$\Lambda_{A \to B} = \sqrt[4]{\frac{2E_{sl}h^3 t}{3\,q_n}},\tag{A1}$$

and between stages B and C (full contact),

$$\Lambda_{B \to C} = \sqrt{3}\,\Lambda_{A \to B}.\tag{A2}$$

[Figure]

**Figure A1.** Comparison of unsupported length and stages between Benedetti et al. (2019) (gray), the present work (black), and the FEA reference solution (circles). Material properties and dimensions are given in **??**.

The assumption of a rigid weak layer in these equations results in significantly larger unsupported lengths compared to both the present model and the finite-element reference model (Fig. A1). Additional details in the model assumptions are listed in Table A1.

**Table A1.** Comparison between Benedetti et al. (2019) and the present model

|  | **Benedetti et al.** | **Present Model** |
|---|---|---|
| **Slab kinematics** | Homogeneous Euler–Bernoulli beam theory | First-order shear deformation theory to account for short beams |
| **Slab layering** | Not accounted for | Laminated plate theory, capturing bending–extension coupling and the influence of slab layering on bending and extensional stiffnesses |
| **Weak layer compliance** | Not accounted for | Weak layer compliance of intact weak layer as well as in touchdown domain considered via a Winkler foundation in the model |
| **Stress analysis** | Rigid beam segment assumption, leading to linear stress distribution along the length | Weak interface solution of shear and normal stresses based on the set of differential equations of the slab deformation (bending deformation, shear deformation, and axial extension). Stress results validated with numerical references models. |
| **Fracture mechanics analyses** | None | Energy release rate of cracks within the weak interface obtained with general mode I and mode II equations for weak interface models. Energy release rate results validated with numerical reference models. |
| **Stages** | | |
| **Stage I / Stage A** | Bending of slab | Bending of slab and weak layer deformation, energy release rate of weak-layer crack propagation |
| **Stage II / Stage B** | No moment introduced at contact point, initial-contact length is a function of (isotropic) slab properties | No moment introduced at contact point, initial-contact length is a function of layered slab and weak-layer properties |
| **Stage III / Stage C** | Slab face is vertical at the contact point (transition from II to III is discontinuous), full-contact length is a function of (isotropic) slab properties | Moments introduced by the slab resting on the collapsed weak layer are represented by rotational springs, whose stiffness is obtained from a full slab deformation analysis by the principle of virtual forces, full-contact length is a function of layered slab and weak-layer properties, energy release rate of weak-layer crack propagation |

**Remark 2.2** *The response to my question regarding stress is disappointing. The authors begin by suggesting that it is common for some people to misunderstand basic mechanics, followed by a lengthy discussion on how stresses at crack tips in elastic continuum-based numerical methods are infinite, in contrast to analytical interface-based models. This is a well-known fact, and I believe there is no need to elaborate on it. Additionally, as the authors are surely aware, plasticity is a simple and effective option (among others) that addresses this issue in continuum-based models, providing a realistic way to simulate the fracture*

*process zone. However, the authors do not employ such a model here. In their approach, as in many previous analytical models,*
35 *finite stress values are obtained at the crack tip, which are crucial for calculating the energy release rate (stress^2 divided by constant stiffness). I feel the sentence: "However, due to their limited applicability in the analysis of fracture phenomena, we chose not to present details of their calculation." very surprising as their calculation is crucial to this paper. Therefore, I would like to reiterate my request: please provide the formula used to calculate this important quantity, as it underpins the entire paper. This could be included in an appendix or supplement. Additionally, please provide the metrics $\tau c/\tau s$ and $\sigma c/\sigma s$, as they*
40 *would allow researchers using similar models to compare results more effectively.*

Thanks for raising this point. We will make it more clear in the revised document. The core point is that the energy release rate is in general not a function of the local stresses but of the change of the total potential $\mathcal{G} = -d\Pi/dA$ at change of crack length. Only in elastic interface models with a simplified continuum in the interface this computation is reduced to Krenk's formula (eq. 11 in the manuscript). In modeling real-world phenomena, all approaches rely on certain assumptions, some of
45 which can lead to unphysical results. A well-known example is beam theory, a fundamental concept in structural mechanics. While the constitutive equation for shear deformation predicts a constant shear stress distribution through the cross-section of a beam, this is incorrect. Engineers account for this by determining shear stresses from local equilibrium conditions, which yield the correct quadratic distribution. A similar principle applies in the present case. Finite crack-tip stresses do not accurately represent the underlying physics of the crack problem. Recognizing this, we do not use stresses to model failure but instead
50 focus on back-calculating the relevant governing properties of the physical process—analogous to how civil engineers avoid assuming constant shear stresses when designing structures. In this context, the energy release rate (ERR) is the fundamental quantity governing crack growth. The presence of stresses in the equation for energy release rate is a consequence of the model assumptions, particularly the weak interface. The derivation of ERR remains consistent across various approaches, including Krenk's method, calculus of variations, and J-integral evaluations. Given this well-established theoretical foundation,
55 our choice to use energy release rate rather than stress-based criteria is both justified and necessary for accurately describing the physical process under investigation. The stress analysis is grounded in the local deformation of the slab above the weak layer. Therefore, the ODEs of the slab deformation are governing the stress solution. We have made this more clear in the paper. We have also added this in the comparison in the supplementary material.

**Remark 2.3** *This comment arose after a second reading of the paper. Specifically, the fact that the ratio Gc/Gs remains*
60 *below one (indicating that the energy release rate exceeds toughness) calls into question the conclusion that crack arrest is not induced by spatial variability. While this may be true (though yet to be fully explained), the findings of the paper, particularly the above point, prevent the authors from definitively concluding this. Spatial variability, especially in weak layer strength, could still explain why Gc/Gs < 1 and in the absence of other explanation, this seems to be a reasonable conclusion. I believe this is a crucial point that warrants further discussion. In fact, the DEM simulations conducted by the authors, or other recently*
65 *developed continuum models (e.g., MPM or FEM), could help to either support or challenge this conclusion.*

We agree that spatial variability might be an important point to understand conditions of crack arrest. We will study this in future works. We have rephrased the paragraph in the manuscript.

**References**

Benedetti, L., Gaume, J., and Fischer, J.-T.: A mechanically-based model of snow slab and weak layer fracture in the Propagation Saw Test, International Journal of Solids and Structures, 158, 1–20, https://doi.org/10.1016/j.ijsolstr.2017.12.033, https://linkinghub.elsevier.com/retrieve/pii/S0020768317305358, 2019.

70